# Coded Sequential Matrix Multiplication For Straggler Mitigation

**M. Nikhil Krishnan**
University of Toronto
nikhilkrishnan.m@gmail.com

**Erfan Hosseini**
University of Toronto
ehosseini2108@gmail.com

**Ashish Khisti**
University of Toronto
akhisti@ece.utoronto.ca

## Abstract

In this work, we consider a sequence of $J$ matrix multiplication jobs which needs to be distributed by a master across multiple worker nodes. For $i \in \{1, 2, \ldots, J\}$, job-$i$ begins in round-$i$ and has to be completed by round-$(i + T)$. Previous works consider only the special case of $T = 0$ and focus on coding across workers. We propose here two schemes with $T > 0$, which feature coding across workers as well as the dimension of time. Our first scheme is a modification of the polynomial coding scheme introduced by Yu et al. and places no assumptions on the straggler model. Exploitation of the temporal dimension helps the scheme handle a larger set of straggler patterns than the polynomial coding scheme, for a given computational load per worker per round. The second scheme assumes a particular straggler model to further improve performance (in terms of encoding/decoding complexity). We develop theoretical results establishing (i) optimality of our proposed schemes for a certain class of straggler patterns and (ii) improved performance for the case of i.i.d. stragglers. These are further validated by experiments, where we implement our schemes to train neural networks.

## 1 Introduction

The sheer scale of data in present day applications necessitates distributing the computation across multiple nodes (will also be referred to as *workers*). One of the key issues faced in distributing the computation across multiple workers is that slow workers (will be referred to as *stragglers*) act as bottlenecks. A naive approach for straggler mitigation is to replicate computation across multiple workers. Clearly, this is wasteful of resources and calls for the need of a systematic approach to introduce redundancy in computational systems. It is in this context that coded computation has been developed, with the goal of introducing redundancy in computation in a resource-efficient manner.

In this paper, we focus on coded computation for distributing matrix multiplication, which is a key building block for linear regression, principal component analysis, training of deep neural networks etc. The idea of introducing redundancy in matrix operations appears in an early paper by Huang and Abraham [1]. The paper [1] considers a multi-processor system and proposes a product-code-based coding scheme to detect and correct errors caused within a single processor. In the distributed matrix multiplication setting, the use of coded computation to provide resiliency against stragglers, has been explored initially in [2]. Coded computation for distributed matrix multiplication (or simply, coded matrix multiplication) has since been actively pursued in the literature. For instance, see [3–15] and references therein. In [16], the authors provide a survey on coded matrix multiplication.

We consider a new setting where a stream of matrix multiplication jobs, indexed by $i \in \{1, 2, \ldots, J\}$, must be finished in a sequential manner. The processing of job-$i$ is initiated in round-$i$, and must be completed by round-$(i + T)$, where $T \geq 0$ will be referred to as the delay parameter. Previous works have exclusively focused on the case, $T = 0$, which is essentially a one-shot setting, where coding can only be done across workers. In contrast, our model enables us to perform coding across both workers and time. We propose two new coded matrix multiplication schemes that exploit these dimensions, illustrate significant gains over previously proposed schemes via theoretical analysis, and present improvements in a practical application of training deep neural networks.

## 2 System model and summary of results

In this section, we present our system model and discuss how our approach differs from existing coded matrix multiplication techniques. For integers $a, b$, let $[a : b] \triangleq \{i \mid a \leq i \leq b\}$.

### 2.1 System model

We consider a distributed system consisting of a master and $P$ workers. Master has to distribute multiplication of a sequence of $J$ pairs of matrices $(X(1), Y(1)), (X(2), Y(2)), \ldots, (X(J), Y(J))$. For $i \in [1 : J]$, we have $X(i) \in \mathbb{R}^{m \times n}$ and $Y(i) \in \mathbb{R}^{n \times p}$. The process of multiplication of matrices $X(i)$ and $Y(i)$ will be referred to as *job-$i$*. The tuple $(X(i), Y(i))$ will be termed as the *input matrix-pair* for job-$i$. The matrices $X(i)$ and $Y(i)$ will be referred to as *input matrices* for job-$i$. The matrix product $X(i) * Y(i)$ will be referred to as the *result* of job-$i$. The master operates based on a certain concept of *rounds* and it takes $J + T$ rounds to complete processing the sequence of $J$ jobs. Here, $T$ is a system parameter, which takes non-negative integer values as discussed below. The time taken to complete each round will depend on the processing speed of the workers. The goal of the master is to finish processing all the $J$ jobs as quickly as possible (in terms of time in seconds).

For $t \in [1 : J + T]$ and $j \in [1 : P]$, in the beginning of round-$t$, master will generate $2\ell_t$ matrices $\{\tilde{X}_j(t; l)\}_{l=1}^{\ell_t}$, $\{\tilde{Y}_j(t; l)\}_{l=1}^{\ell_t}$ and communicate them to each worker-$j$. These matrices $\tilde{X}_j(t; l) \in \mathbb{R}^{\frac{m}{x} \times \frac{n}{z}}$, $\tilde{Y}_j(t; l) \in \mathbb{R}^{\frac{n}{z} \times \frac{p}{y}}$ are arbitrary functions of $\{X(i')\}_{i'=1}^{t}$ and $\{Y(i')\}_{i'=1}^{t}$, respectively. The generation step of these matrices is referred to as *encoding*.

In round-$t$, each worker-$j$ attempts to compute the matrix products $\tilde{X}_j(t; 1) * \tilde{Y}_j(t; 1)$, $\tilde{X}_j(t; 2) * \tilde{Y}_j(t; 2), \ldots, \tilde{X}_j(t; \ell_t) * \tilde{Y}_j(t; \ell_t)$, one after another, in that order. For $l \in [1 : \ell_t]$, the process of multiplication of $\tilde{X}_j(t; l)$ and $\tilde{Y}_j(t; l)$ will be referred to as the *l-th mini-task of worker-$j$ in round-$t$*. The matrices $\tilde{X}_j(t; l)$, $\tilde{Y}_j(t; l)$ will be referred to as *input matrices* for the mini-task. Also, $(\tilde{X}_j(t; l), \tilde{Y}_j(t; l))$ will be referred to as the *input matrix-pair* for this mini-task. The matrix product $\tilde{X}_j(t; l) * \tilde{Y}_j(t; l)$ will be termed the *result* of the mini-task. The result of each mini-task will be communicated to the master as soon as it is ready. When master advances to round-$(t + 1)$, if there are pending mini-tasks in a worker-$j$, it will get canceled. We refer to these canceled mini-tasks as *failed* mini-tasks. The $\ell_t$ mini-tasks assigned by the master to worker-$j$ in round-$t$ will be collectively referred to as a *task*.

Note that by definition, the input matrix-pair $(\tilde{X}_j(t; l), \tilde{Y}_j(t; l))$ for any mini-task assigned to a worker-$j$ in round-$t$, is a function of input matrices for jobs in the range $[1 : t]$. Hence, the workers potentially 'work on' job-$t$ only from round-$t$ onwards. The master has to compute $X(t) * Y(t)$ by the end of round-$(t + T)$, or sooner, using results of non-canceled mini-tasks from all the workers since round-$t$. This step will be referred to as *decoding* (of job-$t$). The parameter $T$ will be naturally referred to as *delay*.

We do not assume that all the input matrix-pairs for the stream of $J$ jobs are available at once to the master. For instance, if these $J$ jobs are to be performed as a part of an iterative algorithm, the input matrices for some job-$i$ might depend on the result of some job-$i'$, $i' < i$. The delay parameter $T$ plays an important role in streamlining processing of jobs under such situations. For instance, a delay of $T \leq i - i' - 1$ immediately helps us manage the dependency between result of job-$i'$ and input matrices for job-$i$. As we will see later in Section 5, such jobs naturally arise during the training of deep neural networks.

*Identification of stragglers*: Stragglers in each round are identified with the help of a tolerance parameter $\mu \geq 0$. Let $\tau(t)$ be the time (in seconds) taken by the fastest worker in round-$t$ to finish all the $\ell_t$ mini-tasks assigned to it and return the results to master. The master waits for $\mu\tau(t)$ more

seconds after it receives the mini-task results from the fastest worker. Any worker who has not finished its $\ell_t$ mini-tasks within this time will be deemed a straggler. Each round-$t$ will thus have a duration of at least $(1 + \mu)\tau(t)$ seconds. The exact manner in which master decides on when to advance from round-$t$ to round-$(t + 1)$ will depend on the coding schemes discussed later.

*Normalized load*: Multiplication of the two matrices $X(t)$ and $Y(t)$ requires $O(mnp)$ floating point operations (assuming the naive matrix multiplication algorithm). Motivated by this, we say that *job load* is $mnp$. Similarly, for each task in round-$t$ consisting of $\ell_t$ matrix multiplications, we have a load of $\ell_t \frac{mnp}{xyz}$. The *normalized task load* in round-$t$ is the ratio $L(t) \triangleq \frac{\ell_t \frac{mnp}{xyz}}{mnp} = \frac{\ell_t}{xyz}$.

*Motivation for coding across time* $(T > 0)$: Existing coded matrix multiplication schemes in the literature implicitly assume $T = 0$, where job-$i$ has to be finished in round-$i$ itself. As a representative scenario, consider the application of *polynomial code* [3] for our setting. In the beginning of round-$i$ $(i \in [1 : J])$, master generates two matrices $\{\tilde{X}_j(i), \tilde{Y}_j(i)\}$ for each worker-$j$ $(j \in [1 : P])$, where $\tilde{X}_j(i) \in \mathbb{R}^{\frac{m}{x} \times n}, \tilde{Y}_j(i) \in \mathbb{R}^{n \times \frac{p}{y}}$ are some functions of $X(i)$ and $Y(i)$, respectively. Each worker-$j$ will attempt to compute the matrix product $\tilde{X}_j(i) * \tilde{Y}_j(i)$ and return this result to the master. As soon as master receives results from some $xy < P$ workers, the master will be able to compute $X(i) * Y(i)$ (via a decoding step). The pending computations being performed by the remaining $S \triangleq (P - xy)$ workers will be canceled and master will enter the next round. Clearly, the system is resilient to $S$ stragglers in each round-$i$, which points to a *persistent straggler pattern* across time. When $T > 0$, master spreads processing of each job across $T + 1$ consecutive rounds. Thus, introduction of the delay parameter $T > 0$ essentially indicates coding across time, which (as we will see later) makes the system robust against more variety of straggler patterns. Interested reader is referred to Section 2.1 of the supplementary material for a motivating example.

*Comparison with streaming codes*: Streaming codes [17] are packet-level forward error correcting codes which enable reliable, high throughput, low-latency communication. Under the streaming code setting, each packet sent in time slot $t$ has to be recovered by time $t + T$. Thus, with regard to the presence of a delay parameter $T$, the framework that we pursue in the paper has some resemblance to the streaming code setting. However, there are fundamental differences in the two approaches, because of which, streaming code constructions do not seem to be applicable to our setting. For instance, consider the streaming code toy example, where packets $p_1, p_2, p_1 + p_2$ are transmitted in time 1, 2 and 3, respectively (any lost packet can be recovered here with a delay of at most 2). Extending this to the matrix multiplication setting, suppose a worker computes $A_1 B_1, A_2 B_2$ and $A_1 B_1 + A_2 B_2$ in successive rounds. This scheme is sub-optimal as $A_1 B_1 + A_2 B_2$ involves 2 matrix multiplications.

## 2.2 Summary of results

- We introduce the problem of coded sequential matrix multiplication, where a stream of matrix multiplication jobs must be completed in a sequential manner. In contrast to the one-shot setting studied in the literature, our setting enables us to take advantage of coding across temporal dimension, as well as across workers.

- In Section 3, we present two new coding schemes. Our first scheme, *diagonally interleaved polynomial (DIP) code* is a natural extension of polynomial code [3] to the sequential setting. Our second scheme, *improved diagonally interleaved polynomial (IDIP) code* improves upon DIP coding in terms of encoding and decoding complexity, for a class of straggler patterns. We present theoretical analysis to illustrate the advantage of our proposed schemes over baseline schemes.

- We present an application of our framework to training deep neural networks in Section 5. Our simulations indicate significant gains when the stragglers are sampled from two statistical models; (i) i.i.d. model and (ii) Fritchman model.

**Remark 2.1** We note that regardless of the actual straggler patterns, our schemes are designed such that job-$i$ will be finished by round-$(i + T)$. The master node ensures this by waiting for stragglers to return mini-task results in certain rounds, if needed, before proceeding to the next round.

**Remark 2.2** For simplicity in exposition, we set $z = 1$ throughout the paper. It is to be noted however that by appropriately modifying the existing coding schemes appearing in works such as [14, 15], the coding scheme presented in Section 3.1 can be generalized to include $z > 1$ case.

# 3 Coded sequential matrix multiplication schemes

## 3.1 Diagonally interleaved polynomial (DIP) coding scheme

In this scheme, in addition to the parameters $T, x, y, z = 1, \mu$ introduced in Section 2.1, we have a hyperparameter $\lambda \in [1 : P]$. Let rows of each $X(i) \in \mathbb{R}^{m \times n}$ (similarly, columns of each $Y(i) \in \mathbb{R}^{n \times p}$), $i \in [-T + 1 : J + T]$, be divided into $x$ submatrices (similarly, $y$ submatrices) as follows:

$$X(i) \triangleq \begin{bmatrix} X(i;1) \\ X(i;2) \\ \vdots \\ X(i;x) \end{bmatrix}, \quad Y(i) \triangleq \begin{bmatrix} Y(i;1) & Y(i;2) & \cdots & Y(i;y) \end{bmatrix},$$

where $X(i;i') \in \mathbb{R}^{\frac{m}{x} \times n}$, $Y(i;j') \in \mathbb{R}^{n \times \frac{p}{y}}$ for $i' \in [1 : x], j' \in [1 : y]$. The submatrices $\{X(i;i')\}$ and $\{Y(i;j')\}$ will be referred to as *subchunks* of $X(i)$ and $Y(i)$, respectively. We also set $X(l') \triangleq \mathbf{0}_{m \times n}, Y(l') \triangleq \mathbf{0}_{n \times p}$, whenever $l' \notin [1 : J]$ (jobs 1 to $J$ are non-trivial, the rest are trivial jobs defined for consistency in notation). We define polynomials $\mathcal{X}_i(\Theta) \triangleq \sum_{i'=1}^{x} X(i;i')\Theta^{y(i'-1)}, \mathcal{Y}_i(\Theta) \triangleq \sum_{j'=1}^{y} Y(i;j')\Theta^{j'-1}$. Note that $\mathcal{X}_i(\Theta) * \mathcal{Y}_i(\Theta)$ takes the form: $\sum_{i' \in [1:x], j' \in [1:y]} [X(i;i') * Y(i;j')]\Theta^{y(i'-1)+j'-1}$, which is a polynomial of degree $xy - 1$. The $xy$ coefficients of this polynomial is precisely given by the set $\chi_i \triangleq \{X(i;i') * Y(i;j')\}_{i' \in [1:x], j' \in [1:y]}$. The matrix product $X(i) * Y(i)$ can be obtained by simply arranging the $xy$ elements of $\chi_i$ in the form of an $m \times p$ matrix. Since all the coefficients of a degree-$(xy - 1)$ polynomial can be retrieved from evaluations at $xy$ distinct points in $\mathbb{R}$, master can essentially compute $X(i) * Y(i)$ from $xy$ evaluations of the polynomial $\mathcal{X}_i(\Theta) * \mathcal{Y}_i(\Theta)$.

In round-$t$ ($t \in [1 : J + T]$), master assigns $\ell_t \in [T : T + \lceil \frac{xy}{\lambda} \rceil]$ mini-tasks to each worker. The exact value of $\ell_t$ is chosen by the master based on the history of mini-task results it has received in the previous rounds. For $l \in [1 : \ell_t]$, the $l$-th mini-task assigned to worker-$j$ ($j \in [1 : P]$) in round-$t$ involves multiplication of two input matrices $\tilde{X}_j(t;l), \tilde{Y}_j(t;l)$. For $l' \in [1 : T]$, $\tilde{X}_j(t;l')$ and $\tilde{Y}_j(t;l')$ are obtained as the evaluations $\mathcal{X}_{t-l'+1}(\Theta)|_{\Theta=\theta_{j,t,l'}}$ and $\mathcal{Y}_{t-l'+1}(\Theta)|_{\Theta=\theta_{j,t,l'}}$, respectively, at some $\theta_{j,t,l'} \in \mathbb{R}$. Naturally, we can say that the $l'$-th mini-task assigned to worker-$j$ in round-$t$ *corresponds to* job-$(t - l' + 1)$. For $l'' \in [T + 1 : \ell_t]$, the $l''$-th mini-task assigned to worker-$j$ in round-$t$ corresponds to job-$(t - T)$.

In Algorithm 1, we formally describe how the master selects each input matrix-pair $(\tilde{X}_j(t;l), \tilde{Y}_j(t;l))$ provided to worker-$j$ in the beginning of round-$t$, as per the DIP coding scheme. For consistency, we assume that job-$i'$ is finished and $xy$ mini-task results corresponding to job-$i'$ are received by the master by default, whenever $i' \notin [1 : J]$.

Let $\tau(t)$ denote the time (in seconds) taken by the fastest worker in round-$t$ to return all its $\ell_t = T + \lceil \frac{xy-\gamma_t}{\lambda} \rceil$ mini-task results to the master. The master waits for $\mu\tau(t)$ more seconds. If $xy - \gamma_t$ mini-task results corresponding to job-$(t - T)$ are received by the master in round-$t$, within this time, it advances to round-$(t + 1)$. If not, as job-$(t - T)$ has to be finished in round-$t$, the master will wait until it receives $xy - \gamma_t$ mini-task results corresponding to job-$(t - T)$. Note that as master assigns $\lceil \frac{xy-\gamma_t}{\lambda} \rceil$ mini-tasks corresponding to job-$(t - T)$ to each worker, even if $P - \lambda$ workers do not return any mini-task results corresponding to job-$(t - T)$, the master can still finish job-$(t - T)$ as soon as it gets mini-task results corresponding to job-$(t - T)$ from the remaining $\lambda$ workers.

**Remark 3.1** As some of the mini-tasks assigned to workers in round-$t$ can be trivial mini-tasks, the normalized task load in round-$t$ given by $L(t) = \frac{\ell_t}{xyz}$ is a worst-case estimate.

The decoding step to finish job-$i$ in DIP coding scheme involves determining the degree-$(xy - 1)$ polynomial $\mathcal{X}_i(\Theta) * \mathcal{Y}_i(\Theta)$ from $xy$ evaluations. If we impose a straggler model and modify the DIP coding scheme to have a model-dependent mini-task assignment algorithm, it is possible to improve the decoding performance. In the following section, we present such a modification.

---
**Algorithm 1:** Algorithm used by master to assign mini-tasks in the DIP coding scheme
---
**1** **for** $j \in [1:P]$ **do**
**2**    **for** $l \in [1:T]$ **do**
**3**      **if** *job-$(t-l+1)$ is finished* **then**
       `// results of `$xy$` mini-tasks corresponding to job-`$(t-l+1)$
       `already received by master`
**4**        assign a trivial mini-task with input matrix pair $(\mathbf{0}_{\frac{m}{x} \times n}, \mathbf{0}_{n \times \frac{p}{y}})$ as the $l$-th mini-task
       of worker-$j$ in round-$t$ (just as a placeholder, will not require any computation)
**5**      **else**
**6**        generate a random number $\theta \in \mathbb{R}$
**7**        pass the evaluations $\mathcal{X}_{t-l+1}(\Theta)|_{\Theta=\theta}, \mathcal{Y}_{t-l+1}(\Theta)|_{\Theta=\theta}$ to worker-$j$
**8**        assign a new mini-task with the input matrices $\mathcal{X}_{t-l+1}(\Theta)|_{\Theta=\theta}, \mathcal{Y}_{t-l+1}(\Theta)|_{\Theta=\theta}$ as
       the $l$-th mini-task of worker-$j$ in round-$t$
**9**    Let $\gamma_t$ denote the number of results of mini-tasks corresponding to job-$(t-T)$ received in
   previous rounds by master.
**10**    **for** $l \in [T+1 : T + \lceil \frac{xy-\gamma_t}{\lambda} \rceil]$ **do**
**11**      generate a random number $\theta \in \mathbb{R}$
**12**      pass the evaluations $\mathcal{X}_{t-T}(\Theta)|_{\Theta=\theta}, \mathcal{Y}_{t-T}(\Theta)|_{\Theta=\theta}$ to worker-$j$
**13**      assign a new mini-task with the input matrices $\mathcal{X}_{t-T}(\Theta)|_{\Theta=\theta}, \mathcal{Y}_{t-T}(\Theta)|_{\Theta=\theta}$ as the $l$-th
     mini-task of worker-$j$ in round-$t$
---

### 3.2 An improved coding scheme for the arbitrary straggler model

#### 3.2.1 $(N, W)$-arbitrary straggler model

The *arbitrary straggler model* is an extension of the persistent straggler model explored in the existing literature (i.e., $S$ stragglers in each round) to include non-persistent straggler patterns as well. The model is parameterized by $W$ and $N \in [0 : WP - 1]$. For $t \in [1 : J + T]$ and $j \in [1 : P]$, let $S_j(t)$ be an indicator function as defined below:

$$S_j(t) = \begin{cases} 1, & \text{Worker-}j \text{ is a straggler in round-}t, \\ 0, & \text{otherwise.} \end{cases} \quad (1)$$

A sliding window of $W$ consecutive rounds is of the form $W_i \triangleq \{i, i+1, i+2, \ldots, i+W-1\}$ ($i \in [1 : J + T - W + 1]$). Let:

$$\mathcal{S}_j(\mathcal{A}) \triangleq \{t' \in \mathcal{A} \mid S_j(t') = 1\},$$

where $\mathcal{A} \subseteq [1 : J + T]$. i.e., $\mathcal{S}_j(\mathcal{A})$ consists of the rounds in $\mathcal{A}$ for which worker-$j$ is a straggler. The straggler pattern seen by the system in rounds $[a : b] \subseteq [1 : J + T]$, where $b - a + 1 \geq W$, is said to conform to the $(N, W)$-arbitrary straggler model, if $\sum_{j=1}^{P} |\mathcal{S}_j(W_i)| \leq N$ for all $W_i \subseteq [a : b]$. If $b - a + 1 < W$, the straggler pattern seen by the system in rounds $[a : b] \subseteq [1 : J + T]$ is said to conform to the $(N, W)$-arbitrary straggler model, if $\sum_{j=1}^{P} |\mathcal{S}_j([a : b])| \leq N$. Furthermore, if worker-$j$ is a straggler in round-$t$, it will not return results of any of the mini-tasks assigned to it in round-$t$.

#### 3.2.2 Improved diagonally interleaved polynomial (IDIP) coding scheme

We consider the $(N, W)$ arbitrary straggler model (i.e., straggler pattern seen by the system in rounds $[1 : J + T]$ conforms to the $(N, W)$-arbitrary straggler model) and show how the DIP scheme can be improved under this model assumption. Our proposed scheme (referred to as IDIP coding scheme) has two advantages; (i) it introduces a fixed load per worker in each round, while the load in the DIP scheme can vary per round, (ii) it takes advantage of uncoded mini-tasks to reduce the encoding and decoding complexity.

Let rows of each $X(i) \in \mathbb{R}^{m \times n}$ (similarly, columns of each $Y(i) \in \mathbb{R}^{n \times p}$), $i \in [-T + 1 : J + T]$, be divided into $x$ subchunks (similarly, $y$ subchunks) as in Section 3.1. Again, we set $X(l') \triangleq \mathbf{0}_{m \times n}, Y(l') \triangleq \mathbf{0}_{n \times p}$, whenever $l' \notin [1 : J]$, as only jobs 1 to $J$ are non-trivial. Let

$k_N \triangleq PW - N, n_N \triangleq W$. Recall the definition of polynomials $\mathcal{X}_i(\Theta) \triangleq \sum_{i'=1}^{x} X(i; i')\Theta^{y(i'-1)}$, $\mathcal{Y}_i(\Theta) \triangleq \sum_{j'=1}^{y} Y(i; j')\Theta^{j'-1}$, whose coefficients are subchunks of $X(i)$ and $Y(i)$, respectively. Here $x, y$ are chosen so that $xy = k_N$. The scheme operates with a delay $T = n_N - 1$.

For each $i \in [-T+1 : J+T]$, the master keeps an ordered list $\mathcal{U}_i$ of the form:

$$\mathcal{U}_i \triangleq \{(X(i;1), Y(i;1)), (X(i;1), Y(i;2)), \ldots, (X(i;x), Y(i;y-1)), (X(i;x), Y(i;y))\},$$

which consists of $xy$ entries. Each entry in the ordered list $\mathcal{U}_i$ is a 2-tuple composed of two matrices; a subchunk of $X(i)$ and $Y(i)$ each. The ordering of these entries within $\mathcal{U}_i$ is in such a way that multiplication of the two subchunks in the $n'$-th entry, for $1 \le n' \le xy$, gives the coefficient corresponding to $\Theta^{n'-1}$ of the polynomial $\mathcal{X}_i(\Theta) * \mathcal{Y}_i(\Theta)$.

In each round-$t$ ($t \in [1 : J+T]$), the master assigns $\ell = n_N = T+1$ mini-tasks to each worker (some of these mini-tasks can be trivial mini-tasks). For $l \in [1 : \ell]$, the $l$-th mini-task assigned to worker-$j$ ($j \in [1 : P]$) in round-$t$ corresponds to job-$(t-l+1)$ in the following sense. The input matrix-pair $(\tilde{X}_j(t;l), \tilde{Y}_j(t;l))$ for this mini-task will be either (i) *uncoded*: one of the entries of $\mathcal{U}_{t-l+1}$ or else (ii) *coded*: evaluation of $(\mathcal{X}_{t-l+1}(\Theta), \mathcal{Y}_{t-l+1}(\Theta))$ at some $\theta_{j,t,l} \in \mathbb{R}$. Each worker-$j$ processes the $\ell$ mini-tasks one after another and communicates each result to the master.

In Algorithm 2, we describe how master selects the input matrix-pair $(\tilde{X}_j(t;l), \tilde{Y}_j(t;l))$ for the $l$-th mini-task to be assigned to each worker-$j$ in round-$t$. We assume that job-$i'$ is finished, whenever $i' \notin [1 : J]$. Furthermore, when $l = 1$ in the outermost loop of Algorithm 2, $\mathcal{J}$ in line-5 becomes the empty set, as mini-tasks are indexed in the range $[1 : \ell]$.

*Improved encoding and decoding performance*: Algorithm 2 is designed such that all uncoded mini-tasks assigned by master will succeed in the $(N, W)$ straggler model. Let $u_i$ indicate the number of uncoded mini-tasks corresponding to job-$i$. In the end of round-$(i+T)$, master is guaranteed to have access to results of; (i) $u_i$ uncoded mini-tasks corresponding to job-$i$ (these $u_i$ mini-tasks have the first $u_i$ entries of $\mathcal{U}_i$ as their input matrices) and (ii) at least $(k_N - u_i)$ coded mini-tasks (these are evaluations of the degree $k_N - 1$ polynomial $\mathcal{X}_i(\Theta) * \mathcal{Y}_i(\Theta)$ at distinct non-zero points). Note that the results of the $u_i$ uncoded mini-tasks, $X(i;1) * Y(i;1), X(i;1) * Y(i;2), \ldots$ are precisely the coefficients of $\Theta^0, \Theta, \ldots, \Theta^{u_i-1}$ in $\mathcal{X}_i(\Theta) * \mathcal{Y}_i(\Theta)$. Owing to the knowledge of these coefficients, any given evaluation of $\mathcal{X}_i(\Theta) * \mathcal{Y}_i(\Theta)$ at some $\theta \ne 0$, can be simplified to evaluation of a polynomial of degree $(k_N - u_i - 1)$ (whose coefficient of $\Theta^l$ is precisely the coefficient of $\Theta^{l+u_i}$ in $\mathcal{X}_i(\Theta)*\mathcal{Y}_i(\Theta)$) at $\theta \ne 0$. In order to finish job-$i$, master determines coefficients of this polynomial via a decoding step which effectively corresponds to inverting a $(k_N - u_i) \times (k_N - u_i)$ Vandermonde matrix. In comparison, in the DIP scheme with parameters $x, y$ such that $xy = k_N$, there are no uncoded mini-tasks and hence, the master deals with a larger, $k_N \times k_N$ Vandermonde matrix. As condition number of an $n \times n$ Vandermonde matrix is known to increase exponentially in $n$ [18] (which makes it increasingly susceptible to numerical errors), it is better to have lower values for $n$. Owing to the presence of uncoded mini-tasks, the encoding complexity of IDIP will also be clearly smaller.

*How to handle a straggler pattern not conforming to the straggler model?*: If the master is in round-$t$, it has the history of stragglers in rounds $[1 : t-1]$. After $(1+\mu)\tau(t)$ seconds into round-$t$, master gets the straggler pattern in round-$t$. If the straggler pattern in rounds $[1 : t]$ conform to the model assumption, the master advances to round-$(t+1)$. Otherwise, master will wait for a few more workers to complete their mini-tasks, mark them as non-stragglers and reassess the straggler pattern. Round-$t$ will finish when a straggler pattern conforming to the model assumption is generated.

### 3.3 Optimality of DIP, IDIP coding schemes under arbitrary straggler model

In the theorem below, we provide a lower bound for worst-case task load of any $z = 1$ scheme under the system model discussed in Section 2.1, with respect to the arbitrary straggler model.

**Theorem 3.1 (Worst-case load for arbitrary straggler model)** *Let $J \to \infty$ and $T < \infty$. The worst-case normalized task load $L$ under the $(N, W)$-arbitrary straggler model is lower bounded as:*

$$L \ge L^* = \frac{1}{P - \frac{N}{W}}. \tag{2}$$

Under the $(N, W)$-arbitrary straggler model, both DIP (for the choice of parameters $xy = PW - N, \lambda = P, T = W - 1$) and IDIP coding schemes can be shown to provide an optimal worst-case normalized task load which matches (2).

**Algorithm 2:** Algorithm used by master to assign mini-tasks in the IDIP coding scheme

---

**1** **for** $l \in [1:\ell]$ **do**

**2**     **if** *job-$(t - l + 1)$ is finished* **then**

**3**        assign a trivial mini-task as the $l$-th mini-task of every worker in round-$t$

**4**     **else**

**5**        Let $\mathcal{J} = \{j \in [1:P] \mid (l-1)$-th mini-task assigned to worker-$j$ in round-$(t-1)$ has failed$\}$

**6**        **for** $j \in \mathcal{J}$ **do**

**7**           reassign the failed $(l-1)$-th mini-task of worker-$j$ as the $l$-th mini-task of worker-$j$ in round-$t$

**8**        **for** $j \in [1:P] \setminus \mathcal{J}$ **do**

**9**           **if** *number of remaining unassigned entries of $\mathcal{U}_{t-l+1} < (P - |\mathcal{J}|)$* **then**

**10**              generate a random number $\theta \in \mathbb{R} \setminus \{0\}$

**11**              pass the evaluations $\mathcal{X}_{t-l+1}(\Theta)|_{\Theta=\theta}, \mathcal{Y}_{t-l+1}(\Theta)|_{\Theta=\theta}$ to worker-$j$

**12**              assign a new mini-task with the input matrices $\mathcal{X}_{t-l+1}(\Theta)|_{\Theta=\theta}, \mathcal{Y}_{t-l+1}(\Theta)|_{\Theta=\theta}$ as the $l$-th mini-task of worker-$j$ in round-$t$

**13**           **else if** *worker-$j$ cannot be a straggler during at least one of the rounds in $[t : t + n_N - l]$, due to the straggler model* **then**

**14**              pass the two subchunks in the first previously unassigned entry of $\mathcal{U}_{t-l+1}$ to worker-$j$

**15**              assign the $l$-th mini-task of worker-$j$ in round-$t$, with this entry as the input matrix-pair

**16**           **else**

**17**              generate a random number $\theta \in \mathbb{R} \setminus \{0\}$

**18**              pass the evaluations $\mathcal{X}_{t-l+1}(\Theta)|_{\Theta=\theta}, \mathcal{Y}_{t-l+1}(\Theta)|_{\Theta=\theta}$ to worker-$j$

**19**              assign a new mini-task with the input matrices $\mathcal{X}_{t-l+1}(\Theta)|_{\Theta=\theta}, \mathcal{Y}_{t-l+1}(\Theta)|_{\Theta=\theta}$ as the $l$-th mini-task of worker-$j$ in round-$t$

---

## 4 Numerical results

In this section, we analytically compare the performance of our two schemes with the polynomial coding scheme. Proofs are provided in Section 4 of the supplementary material. We consider in this section, a probabilistic, i.i.d. straggler model, i.e., any worker in any given round will be a straggler with probability $\delta$. We make the simplifying assumption that for a normalized task load $L(t)$ in round-$t$, a non-straggler takes $\tau L(t)$ seconds to finish the task. Conversely, a straggler takes $\alpha \tau L(t)$ seconds to finish the task ($\alpha > 1$). As randomness in this system is occurring only due to the straggler model, we set $\mu = 0$ (the parameter used by master to detect stragglers in each round). Let $R_J$ denote the time required to complete $J$ jobs. We have $\hat{R} \triangleq \lim_{J \to \infty} \frac{\mathbb{E}[R_J]}{J\tau}$.

**Polynomial coding scheme** Let the code be resilient against $S < P$ stragglers. We have $\hat{R}^{\text{poly}}_{\alpha,S,P,\delta} = \frac{\alpha p^{\text{poly}}_{S,P,\delta} + (1 - p^{\text{poly}}_{S,P,\delta})}{P-S}$, where $p^{\text{poly}}_{S,P,\delta} \triangleq \sum_{i=S+1}^{P} \binom{P}{i} \delta^i (1-\delta)^{P-i}$. Note that $\hat{R}^{\text{poly}}_{\alpha,S,P,\delta}$ essentially is the product of normalized task load $\frac{1}{P-S}$ and expected scaling in round duration due to stragglers. When $S = 0$, the scheme becomes equivalent to the uncoded scheme, i.e., $\hat{R}^{\text{uncoded}}_{\alpha,P,\delta} = \hat{R}^{\text{poly}}_{\alpha,0,P,\delta} = \frac{\alpha(1-(1-\delta)^P) + (1-\delta)^P}{P}$.

**IDIP coding scheme** Here, we study the performance of the IDIP coding scheme designed for the $(N, W)$-arbitrary straggler model, although the stragglers are sampled from an i.i.d. model. We have $\hat{R}^{\text{IDIP}}_{\alpha,N,W,P,\delta} \leq \frac{\alpha p^{\text{IDIP}}_{N,W,P,\delta} + (1 - p^{\text{IDIP}}_{N,W,P,\delta})}{P - \frac{N}{W}}$, where $p^{\text{IDIP}}_{N,W,P,\delta} \triangleq \sum_{i=N+1}^{PW} \binom{PW}{i} \delta^i (1-\delta)^{PW-i}$. Similar to the earlier schemes, here, $\frac{1}{P - \frac{N}{W}}$ is the normalized task load and $\alpha p^{\text{IDIP}}_{N,W,P,\delta} + (1 - p^{\text{IDIP}}_{N,W,P,\delta})$ is an upper bound on the expected scaling in round duration. In order to compare $\hat{R}^{\text{IDIP}}_{\alpha,N,W,P,\delta}$ with $\hat{R}^{\text{poly}}_{\alpha,S,P,\delta}$, set $N = SW$. It can be shown that $\hat{R}^{\text{IDIP}}_{\alpha,N,W,P,\delta} \leq \hat{R}^{\text{poly}}_{\alpha,S,P,\delta}$.

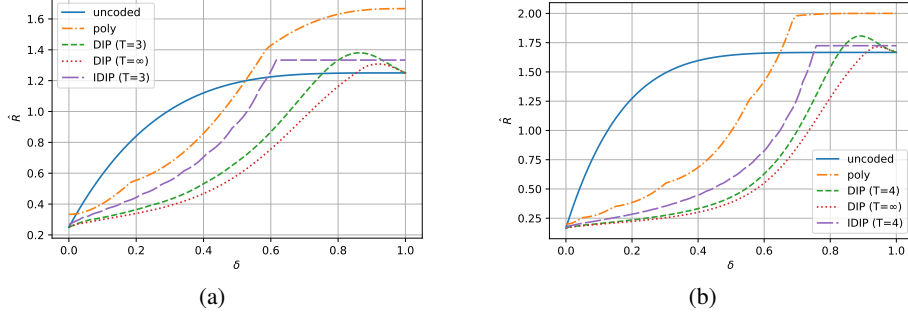

(a)                                    (b)

Figure 1: A plot of $\delta$ vs. $\hat{R}$. In Fig. (a) and Fig. (b), we consider $\{\alpha = 5, P = 4, T = W - 1 = 3\}$ and $\{\alpha = 10, P = 6, T = W - 1 = 4\}$, respectively. For given $\{\alpha, P, \delta\}$, we minimize each of $\hat{R}^{\text{poly}}_{\alpha,S,P,\delta}$, $\hat{R}^{\text{DIP}}_{\alpha,\beta,\infty,\infty,P,\delta}$, $\hat{R}^{\text{DIP}}_{\alpha,\beta,T,\lambda,P,\delta}$, $\hat{R}^{\text{IDIP}}_{\alpha,N,W,P,\delta}$ (upper bound) with respect to $\{S \geq 1\}$, $\{\beta \geq P\}$, $\{\beta \geq P, \lambda\}$ and $\{N \geq 1\}$, respectively.

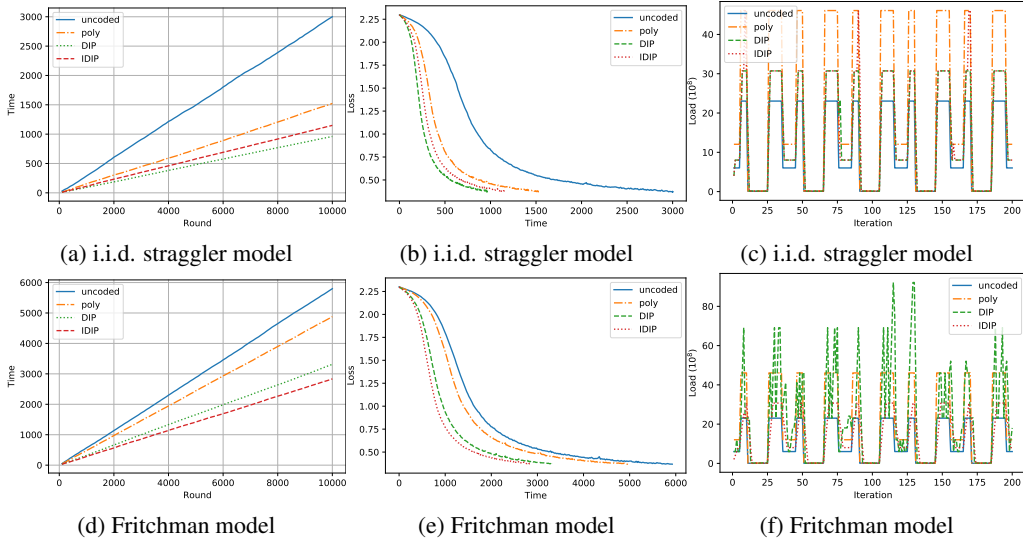

(a) i.i.d. straggler model     (b) i.i.d. straggler model     (c) i.i.d. straggler model

(d) Fritchman model            (e) Fritchman model            (f) Fritchman model

Figure 2: Experimental results for training neural networks. Fig. (a) & (d): round vs. cumulative processing time; Fig. (b) & (e): time vs. test loss for NN model-1; Fig. (c) & (f): round (first 200) vs. task load. The task loads vary across rounds for all schemes, as job loads are not the same across rounds. In particular, the load-variation can be seen to be amplified for the DIP scheme (due to its inherent variable load nature) in Fig. (f).

**DIP coding scheme**   In our discussion here, we focus on the limiting case when $T \to \infty$, where the analysis simplifies to some extent. Note that even in this case, the expected number of rounds required to finish processing a job (denoted by $f_{\beta,P,\delta}$) is a finite quantity. Here $\beta = xy$, is the total number of mini-tasks needed for completing each job. The actual expression for $f_{\beta,P,\delta}$ is derived in the supplementary material. The parameter $\lambda$ is inactive when $T = \infty$ (we simply set $\lambda \triangleq \infty$ to emphasize that it is inactive). It can be shown that: $\hat{R}^{\text{DIP}}_{\alpha,\beta,T=\infty,\lambda=\infty,P,\delta} = \frac{f_{\beta,P,\delta}}{\beta}(\alpha p^{\text{all}}_{\delta,P} + (1 - p^{\text{all}}_{\delta,P}))$, where $p^{\text{all}}_{\delta,P} \triangleq \delta^P$ is the probability that all workers in a round are stragglers. Through numerical evaluation in Fig. 1, it can be observed that DIP scheme outperforms polynomial and IDIP schemes.

## 5   Experimental results

In this section, we evaluate the performance of proposed schemes by training 5 neural network (NN) models concurrently (with learning rates $\{0.1, 0.15, 0.2, 0.25, 0.3\}$). Algorithms are implemented using mpi4py [19] and NumPy on a local university testbed. We use four virtual machines with 8GB of RAM and 4 vCPUs as workers and one more machine with 16GB of RAM and 8 vCPUs as the master. The master distributes jobs, keeps track of stragglers, and decodes the job results. During

the wait-time to obtain results from workers in each round, master performs decoding of results from previous rounds and any necessary encoding for the next round. Master performs encoding and decoding at the same time on multiple vCPUs. Our experiments show that average encoding/decoding times are smaller than average processing times (denoted by $R \triangleq \frac{R_J}{J}$). Consequently, the effect encoding/decoding times have in our experiments is not significant.

Each NN model is fully connected with two hidden layers of size 3000 followed by a ReLU activation. Training is performed for 250 iterations over MNIST dataset with a batch size of 1024 using SGD. We break down each iteration of training into 8 sequential matrix-matrix multiplication jobs (job loads vary across these 8 jobs); 3 and 5 jobs respectively for forward and backward passes. The jobs belonging to the 5 NN models are interleaved so that job-$i$ belongs to model $((i - 1) \bmod 5) + 1$. Hence, input matrices for job-$i$ is dependent on the result of job-$(i - 5)$. It takes $T + 1$ rounds (worst-case) to deliver mini-task results corresponding to each job and then, one additional round for decoding. Thus, $T$ is set to 3. For the 5 NN models, we have in total $J = 250 * 5 * 8 = 10000$ jobs.

We run the experiments based on the i.i.d. straggler model, as well as the Fritchman model, which models presence of stragglers in bursts. In order to simulate stragglers, we make the to-be-stragglers perform their tasks $\alpha = 5$ times. We select best-performing code parameters for each straggler model using a simplified first order simulation. For the i.i.d. model, we set the straggler probability $\delta = 0.3$. The code parameters used are (i) polynomial: $\{S = 2, x = 2, y = 1\}$ (ii) DIP: $\{x = 2, y = 3, T = 3, \lambda = 1, \mu = 0.25\}$ (iii) IDIP: $\{x = 2, y = 3, T = 3, N = 10, \mu = 0.25\}$. For the Fritchman model (details of the model and parameters can be found in Section 5 of the supplementary material), code parameters used are (i) polynomial: $\{S = 2, x = 2, y = 1\}$ (ii) DIP: $\{x = 2, y = 2, T = 3, \lambda = 2, \mu = 0.25\}$ (iii) IDIP: $\{x = 2, y = 6, T = 3, N = 4, \mu = 0.25\}$ (IDIP scheme here is a variant tailored for bursty stragglers, discussed in Section 3.2.3 of the supplementary material). A performance summary is provided in Table 1. Fig. 2 clearly depicts the improvement newly proposed schemes provide over the polynomial coding scheme. DIP, IDIP schemes register reductions of 36% (32%) and 24% (41%), respectively, in the average processing time ($R$) over polynomial codes, for i.i.d. (Fritchman) straggler model. Note that experimental results are in agreement with the numerical results, which assume the i.i.d. straggler model and predict superior performance of DIP scheme over both IDIP and polynomial schemes. For the Fritchman model, IDIP emerges to be the best-performing scheme.

Table 1: Performance summary for two straggler models, in terms of encoding/decoding times and $R$.

(a) i.i.d. model

| Scheme | Enc. time | Dec. time | $R$ |
|---|---|---|---|
| Uncoded | 0 | 0 | 0.3 |
| Poly | 0.076 | 0.077 | 0.152 |
| DIP | 0.091 | 0.082 | 0.096 |
| IDIP | 0.052 | 0.078 | 0.115 |

(b) Fritchman model

| Scheme | Enc. time | Dec. time | $R$ |
|---|---|---|---|
| Uncoded | 0 | 0 | 0.574 |
| Poly | 0.045 | 0.187 | 0.482 |
| DIP | 0.065 | 0.206 | 0.327 |
| IDIP | 0.024 | 0.193 | 0.281 |

**Remark 5.1 (Applicability of the scheme in general)** Even though we discuss the specific application of training multiple NN simultaneously, our framework suits well in any situation where the master is interested in finishing quickly a collection of multiple independent sequences of matrix multiplications (dependencies are permitted within a sequence). For instance, solving multiple systems of linear equations through an iterative algorithm such as the Jacobi method.

## Broader Impact

The new coding schemes that we propose here aim to reduce the cumulative processing time of a sequence of matrix multiplication jobs. This could result in energy savings. Hence, our work has the potential to contribute towards energy initiatives.

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
