[Supplementary Material · 6645_Coded_Seq_Mult_For_Straggler_Mitigation.pdf]

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

 consistency in notation, we additionally define $T$ *trivial jobs*, $\{\text{job-}i'\}_{i' \in [J+1:J+T]}$. For each trivial job-$i'$, the input matrices are both all-zero matrices and the result of job-$i'$, i.e., $\mathbf{0}_{m \times p}$, is known to the master by default. For $t \in [1 : J + T]$ and $j \in [1 : P]$, in the beginning of round-$t$, master will generate $2\ell_t$ matrices $\{\tilde{X}_j(t; l)\}_{l=1}^{\ell_t}$, $\{\tilde{Y}_j(t; l)\}_{l=1}^{\ell_t}$ and communicate them to each worker-$j$. These matrices $\tilde{X}_j(t; l) \in \mathbb{R}^{\frac{m}{x} \times \frac{n}{z}}$, $\tilde{Y}_j(t; l) \in \mathbb{R}^{\frac{n}{z} \times \frac{p}{y}}$ are arbitrary functions of $\{X(i')\}_{i'=1}^{t}$ and $\{Y(i')\}_{i'=1}^{t}$, respectively. The generation step of these matrices is referred to as *encoding*.

In round-$t$, each worker-$j$ attempts to compute the matrix products $\tilde{X}_j(t; 1) * \tilde{Y}_j(t; 1)$, $\tilde{X}_j(t; 2) * \tilde{Y}_j(t; 2), \ldots, \tilde{X}_j(t; \ell_t) * \tilde{Y}_j(t; \ell_t)$, one after another, in that order. For $l \in [1 : \ell_t]$, the process of multiplication of $\tilde{X}_j(t; l)$ and $\tilde{Y}_j(t; l)$ will be referred to as the *l-th mini-task of worker-j in round-t*. The matrices $\tilde{X}_j(t; l), \tilde{Y}_j(t; l)$ will be referred to as *input matrices* for the mini-task. Also, $(\tilde{X}_j(t; l), \tilde{Y}_j(t; l))$ will be referred to as the *input matrix-pair* for this mini-task. The matrix product $\tilde{X}_j(t; l) * \tilde{Y}_j(t; l)$ will be termed the *result* of the mini-task. The result of each mini-task will be communicated to the master as soon as it is ready. When master advances to round-$(t + 1)$, if there are pending mini-tasks in a worker-$j$, it will get canceled. We refer to these canceled mini-tasks as *failed* mini-tasks. The $\ell_t$ mini-tasks assigned by the master to worker-$j$ in round-$t$ will be collectively referred to as a *task*.

Note that by definition, the input matrix-pair $(\tilde{X}_j(t; l), \tilde{Y}_j(t; l))$ for any mini-task assigned to a worker-$j$ in round-$t$, is a function of input matrices for jobs in the range $[1 : t]$. Hence, the workers potentially 'work on' job-$t$ only from round-$t$ onwards. The master has to compute $X(t) * Y(t)$ by the end of round-$(t + T)$, or sooner, using results of non-canceled mini-tasks from all the workers since round-$t$. This step will be referred to as *decoding* (of job-$t$). The parameter $T$ will be naturally referred to as *delay*. In Fig. 1, we provide an overview of the framework that we consider in this paper.

We do not assume that all the input matrix-pairs for the stream of $J$ jobs are available at once to the master. For instance, if these $J$ jobs are to be performed as a part of an iterative algorithm, the input matrices for some job-$i$ might depend on the result of some job-$i'$, $i' < i$. The delay parameter $T$ plays an important role in streamlining processing of jobs under such situations. For instance, a delay of $T \leq i - i' - 1$ immediately helps us manage the dependency between result of job-$i'$ and input matrices for job-$i$. As we will see later in Section 5, such jobs naturally arise during the training of deep neural networks.

Figure 1: An overview of the setting we pursue in the paper. The master needs to distribute and finish the processing of a stream of $J$ matrix multiplication jobs in at most $J + T$ rounds. In the beginning of each round-$t$, master assigns $\ell_t$ mini-tasks to each worker. Each worker attempts to finish one after another, all the mini-tasks assigned to it. In this figure, worker-2 is a potential straggler who lags behind in sending its mini-task results to the master, in comparison to other workers. Through a decoding step, master obtains the result of job-$i$ ($i \in [1 : J]$) from a subset of mini-task results received in rounds $[i : i + T]$.

*Identification of stragglers*: Stragglers in each round are identified with the help of a tolerance parameter $\mu \geq 0$. Let $\tau(t)$ be the time (in seconds) taken by the fastest worker in round-$t$ to finish all the $\ell_t$ mini-tasks assigned to it and return the results to master. The master waits for $\mu\tau(t)$ more seconds after it receives the mini-task results from the fastest worker. Any worker who has not finished its $\ell_t$ mini-tasks within this time will be deemed a straggler. Each round-$t$ will thus have a duration of at least $(1 + \mu)\tau(t)$ seconds. The exact manner in which master decides on when to advance from round-$t$ to round-$(t + 1)$ will depend on the coding schemes discussed later.

*Normalized load*: Multiplication of the two matrices $X(t)$ and $Y(t)$ requires $O(mnp)$ floating point operations (assuming the naive matrix multiplication algorithm). Motivated by this, we say that *job load* is $mnp$. Similarly, for each task in round-$t$ consisting of $\ell_t$ matrix multiplications, we have a load of $\ell_t \frac{mnp}{xyz}$. The *normalized task load* in round-$t$ is the ratio $L(t) \triangleq \frac{\ell_t \frac{mnp}{xyz}}{mnp} = \frac{\ell_t}{xyz}$.

*Why $T > 0$ can help?*: Existing coded matrix multiplication schemes in the literature implicitly assume $T = 0$, where job-$i$ has to be finished in round-$i$ itself. As a representative scenario, consider the application of *polynomial code* [3] for our setting. In the beginning of round-$i$ ($i \in [1 : J]$), master generates two matrices $\{\tilde{X}_j(i), \tilde{Y}_j(i)\}$ for each worker-$j$ ($j \in [1 : P]$), where $\tilde{X}_j(i) \in \mathbb{R}^{\frac{m}{x} \times n}, \tilde{Y}_j(i) \in \mathbb{R}^{n \times \frac{p}{y}}$ are some functions of $X(i)$ and $Y(i)$, respectively. Each worker-$j$ will attempt to compute the matrix product $\tilde{X}_j(i) * \tilde{Y}_j(i)$ and return this result to the master. As soon as master receives results from some $xy < P$ workers, the master will be able to compute $X(i) * Y(i)$ (via a decoding step). The pending computations being performed by the remaining $S \triangleq (P - xy)$ workers will be canceled and master will enter the next round. Clearly, the system is resilient to $S$ stragglers in each round-$i$, which points to a *persistent straggler pattern* across time. Introduction of the delay parameter $T > 0$ essentially indicates coding across time, which can possibly make the system robust against persistent straggler patterns *as well as* non-persistent straggler patterns (for instance, a scenario where there are no stragglers in round-1 and $2S$ stragglers in round-2).

**Motivating example** Consider a distributed system with a master and three workers. In Fig. 2a, which is representative of the existing approaches, we consider the application of a simple parity check code to split each matrix multiplication job into three 'smaller' matrix multiplication jobs (mini-tasks). Master partitions rows of each $X(i)$ into $x = 2$ sub-matrices $\{X(i; l)\}_{l=1}^2$. In each

Figure 2: (a) A system resilient to one straggler in every round (b) a system resilient to two stragglers in every two consecutive rounds. Here, shaded rectangles indicate mini-tasks not returned by stragglers.

round-$i$, $i \in \{1, 2, \ldots, J = 4\}$, master provides to each worker-$j$, $j \in \{1, 2, 3\}$, a pair of matrices $(\tilde{X}_j(i), Y(i))$, where $\tilde{X}_1(i) = X(i;1)$, $\tilde{X}_2(i) = X(i;2)$ and $\tilde{X}_3(i) = X(i;1) + X(i;2)$. Each worker-$j$ computes the matrix product $\tilde{X}_j(i) * Y(i)$. It is straightforward to verify from 2a that the master can compute $X(i) * Y(i)$ from the results returned by any two workers, in every round-$i$. Thus, the system can handle one straggler in every round, which points to a persistent straggler pattern across rounds. Assume that a non-straggling worker (non-straggler) takes $\tau$ seconds to finish the mini-task (with load $\frac{mnp}{2}$) assigned to it, whereas a straggler takes $2\tau$ seconds to finish the same mini-task. For the scenario described in Fig. 2a, rounds 2 & 4 are 'bad', as there are two stragglers in each of these rounds and the system is resilient only to one straggler per round. Thus, master has to wait for stragglers to finish their mini-tasks in these rounds. The approach in Fig. 2a takes $\tau + 2\tau + \tau + 2\tau = 6\tau$ seconds to complete four jobs.

In Fig.2b, the master 'spreads' processing of each job across two consecutive round (thus, there is a delay of one round). Master partitions rows of each $X(i)$ into $x = 4$ sub-matrices $\{X(i;l)\}_{l=1}^4$. In Fig. 2b, $\tilde{X}(i;1)$, $\tilde{X}(i;2)$ are coded matrices so that $\{X(i;l) * Y(i)\}_{l=1}^4$ can be retrieved from any four out of the six matrix products $\{X(i;l) * Y(i)\}_{l=1}^4 \cup \{\tilde{X}(i;1) * Y(i), \tilde{X}(i;2) * Y(i)\}$. As in the previous case, load per worker in each round is $2 * \frac{mnp}{4} = \frac{mnp}{2}$. Moreover, from Fig. 2b, it can be verified that this approach too is resilient to the persistent straggler scenario, i.e., one straggler in each round. However, this approach is also resilient to the scenario of alternating 'good' and 'bad' rounds as depicted in Fig. 2b. The system here only takes $\tau + \tau + \tau + \tau + \tau = 5\tau$ seconds to finish four jobs, despite the delay of one round.

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

 Fig. 3, we illustrate how mini-tasks correspond to different jobs. It can be observed that mini-tasks corresponding to each job are distributed in a near-diagonal manner across rounds. This explains the name "diagonally interleaved polynomial coding scheme".

In Algorithm 1, we formally describe how the master selects each input matrix-pair $(\tilde{X}_j(t;l), \tilde{Y}_j(t;l))$ provided to worker-$j$ in the beginning of round-$t$, as per the DIP coding scheme. For consistency, we assume that job-$i'$ is finished and $xy$ mini-task results corresponding to job-$i'$ are received by the master by default, whenever $i' \notin [1 : J]$.

Let $\tau(t)$ denote the time (in seconds) taken by the fastest worker in round-$t$ to return all its $\ell_t = T + \max(\lceil \frac{xy - \gamma_t}{\lambda} \rceil, 0)$ mini-task results to the master. The master waits for $\mu\tau(t)$ more seconds. If

Figure 3: An illustration of how each mini-task corresponds to a job. Each rectangle here depicts a mini-task. The number inside each rectangle indicates which job does that particular mini-task correspond to. Uncolored rectangles indicate trivial mini-tasks corresponding to trivial jobs, which are just place-holders for notational convenience and do not have to be processed by a worker.

---

**Algorithm 1:** Algorithm used by master to assign mini-tasks in the DIP coding scheme

---

1   **for** $j \in [1 : P]$ **do**
2     **for** $l \in [1 : T]$ **do**
3       **if** *job-$(t - l + 1)$ is finished* **then**
         `// results of` $xy$ `mini-tasks corresponding to job-`$(t-l+1)$
         `already received by master`
4         assign a trivial mini-task with input matrix pair $(\mathbf{0}_{\frac{m}{x} \times n}, \mathbf{0}_{n \times \frac{p}{y}})$ as the $l$-th mini-task of worker-$j$ in round-$t$ (just as a placeholder, will not require any computation)
5       **else**
6         generate a random number $\theta \in \mathbb{R}$
7         pass the evaluations $\mathcal{X}_{t-l+1}(\Theta)|_{\Theta=\theta}, \mathcal{Y}_{t-l+1}(\Theta)|_{\Theta=\theta}$ to worker-$j$
8         assign a new mini-task with the input matrices $\mathcal{X}_{t-l+1}(\Theta)|_{\Theta=\theta}, \mathcal{Y}_{t-l+1}(\Theta)|_{\Theta=\theta}$ as the $l$-th mini-task of worker-$j$ in round-$t$
9     Let $\gamma_t$ denote the number of results of mini-tasks corresponding to job-$(t - T)$ received in previous rounds by master.
10     **for** $l \in [T + 1 : T + \lceil \frac{xy - \gamma_t}{\lambda} \rceil]$ **do**
11       generate a random number $\theta \in \mathbb{R}$
12       pass the evaluations $\mathcal{X}_{t-T}(\Theta)|_{\Theta=\theta}, \mathcal{Y}_{t-T}(\Theta)|_{\Theta=\theta}$ to worker-$j$
13       assign a new mini-task with the input matrices $\mathcal{X}_{t-T}(\Theta)|_{\Theta=\theta}, \mathcal{Y}_{t-T}(\Theta)|_{\Theta=\theta}$ as the $l$-th mini-task of worker-$j$ in round-$t$

---

$\max(xy - \gamma_t, 0)$ mini-task results corresponding to job-$(t - T)$ are received by the master in round-$t$ within this time, it advances to round-$(t + 1)$. If not, as job-$(t - T)$ has to be finished in round-$t$, the master will wait until it receives $\max(xy - \gamma_t, 0)$ mini-task results corresponding to job-$(t - T)$ and then proceed to round-$(t+1)$. If $\gamma_t < xy$, note that master assigns $\lceil \frac{xy - \gamma_t}{\lambda} \rceil$ mini-tasks corresponding to job-$(t - T)$ to each worker. In round-$t$, even if $P - \lambda$ workers do not return any mini-task results corresponding to job-$(t-T)$, the master can still finish job-$(t-T)$ as soon as it gets mini-task results corresponding to job-$(t - T)$ from the remaining $\lambda$ workers.

**Remark 3.1.** As some of the mini-tasks assigned to workers in round-$t$ can be trivial mini-tasks which do not require any processing, the normalized task load in round-$t$ given by $L(t) = \frac{\ell_t}{xyz}$ is a worst-case estimate.

**Remark 3.2.** In the worst-case scenario, (non-trivial) mini-tasks corresponding to a job-$i$ will be present in all the $T + 1$ consecutive rounds $[i : i + T]$. However, to the contrary, if the number of stragglers the system encounters is small, the number of rounds required to process a job could be as small as $\min(\lceil \frac{xy}{P} \rceil, T + 1)$.

The decoding step to finish job-$i$ in DIP coding scheme involves determining the degree-$(xy - 1)$ polynomial $\mathcal{X}_i(\Theta) * \mathcal{Y}_i(\Theta)$ from $xy$ evaluations (polynomial interpolation). If we impose a straggler model and modify the DIP coding scheme to have a model-dependent mini-task assignment algorithm,

Figure 4: A straggler pattern conforming to the $(N = 4, W = 5)$-arbitrary straggler model. In every sliding window of size $W = 5$, there are at most $N = 4$ stragglers in total. Each shaded rectangle indicates a straggler.

it is possible to improve the decoding performance. In the following section, we present such a modification.

## 3.2 An improved coding scheme for specific straggler models

### 3.2.1 $(N, W)$-arbitrary straggler model

The *arbitrary straggler model* is an extension of the persistent straggler model explored in the existing literature (i.e., $S$ stragglers in each round) to include non-persistent straggler patterns as well. The model is parameterized by $W$ and $N \in [0 : WP - 1]$. For $t \in [1 : J + T]$ and $j \in [1 : P]$, let $S_j(t)$ be an indicator function as defined below:

$$S_j(t) = \begin{cases} 1, & \text{Worker-}j \text{ is a straggler in round-}t, \\ 0, & \text{otherwise.} \end{cases} \tag{1}$$

A sliding window of $W$ consecutive rounds is of the form $W_i \triangleq \{i, i + 1, i + 2, \ldots, i + W - 1\}$ ($i \in [1 : J + T - W + 1]$). Let:

$$S_j(\mathcal{A}) \triangleq \{t' \in \mathcal{A} \mid S_j(t') = 1\},$$

where $\mathcal{A} \subseteq [1 : J + T]$. i.e., $S_j(\mathcal{A})$ consists of the rounds in $\mathcal{A}$ for which worker-$j$ is a straggler. The straggler pattern seen by the system in rounds $[a : b] \subseteq [1 : J + T]$, where $b - a + 1 \geq W$, is said to conform to the $(N, W)$-arbitrary straggler model, if $\sum_{j=1}^{P} |S_j(W_i)| \leq N$ for all $W_i \subseteq [a : b]$ (see Fig. 4). If $b - a + 1 < W$, the straggler pattern seen by the system in rounds $[a : b] \subseteq [1 : J + T]$ is said to conform to the $(N, W)$-arbitrary straggler model, if $\sum_{j=1}^{P} |S_j([a : b])| \leq N$. Furthermore, if worker-$j$ is a straggler in round-$t$, it will not return results of any of the mini-tasks assigned to it in round-$t$.

### 3.2.2 $(B, \epsilon, W)$-bursty straggler model

In this deterministic straggler model, which we will refer to as the *bursty straggler model*, we deal with the scenario that a worker can behave as a straggler for several consecutive rounds. The model is parameterized by $W$, $\epsilon \in [0 : P]$ and $B \in [1 : W]$. The configuration $\{\epsilon = P, B = W\}$ is infeasible. For $i \in [1 : J + T - W + 1]$, $j \in [1 : P]$ and $t \in [1 : J + T]$, let $W_i, S_j(t), S_j(.)$ be as defined in Section 3.2.1. Let $\mathcal{B} \subseteq [1 : J + T]$. We have $\mathcal{P}(\mathcal{B}) \triangleq \{j \in [1 : P] \mid S_j(t') = 1 \text{ for some } t' \in \mathcal{B}\}$. For the null-set $\phi$, we set $\max(\phi) = \min(\phi) \triangleq 0$. The straggler pattern seen by the system in rounds $[a : b] \subseteq [1 : J + T]$, where $b - a + 1 \geq W$, is said to conform to the $(B, \epsilon, W)$-bursty straggler model, if it satisfies the two conditions:

- $|\mathcal{P}(W_i)| \leq \epsilon$,
- $\max(S_j(W_i)) - \min(S_j(W_i)) + 1 \leq B$,

for all $W_i \subseteq [a : b]$ and $j \in [1 : P]$. If $b - a + 1 < W$, the straggler pattern seen by the system in rounds $[a : b] \subseteq [1 : J + T]$ is said to conform to the $(B, \epsilon, W)$-bursty straggler model, if it satisfies the two conditions:

- $|\mathcal{P}([a : b])| \leq \epsilon$,

Figure 5: A straggler pattern conforming to the $(B = 2, \epsilon = 2, W = 5)$-bursty straggler model. In every sliding window of size $W = 5$, there are at most $\epsilon = 2$ workers who behave as stragglers for at most $B = 2$ consecutive rounds. Each shaded rectangle indicates a straggler.

- $\max(\mathcal{S}_j([a:b])) - \min(\mathcal{S}_j([a:b])) + 1 \leq B$.

In other words, under the $(B, \epsilon, W)$-bursty straggler model, within any sliding window $W_i$ of size $W$, at most $\epsilon$ out of $P$ workers are permitted to be stragglers and a worker can remain a straggler only for a maximum of $B$ consecutive rounds (see Fig. 5). As in the previous model, a straggler in a round will not return any results.

### 3.2.3 Improved diagonally interleaved polynomial (IDIP) coding scheme

We consider here the $(N, W)$ arbitrary straggler model (i.e., straggler pattern seen by the system in rounds $[1 : J + T]$ conforms to the $(N, W)$-arbitrary straggler model) and show how the DIP scheme can be improved under this model assumption (subsequently, in Remark 3.3, we discuss how the scheme can be extended for the bursty straggler model). Our proposed scheme (referred to as IDIP coding scheme) has two advantages; (i) it introduces a fixed load per worker in each round, while the load in the DIP scheme can vary per round, (ii) it takes advantage of uncoded mini-tasks to reduce the encoding and decoding complexity.

Let rows of each $X(i) \in \mathbb{R}^{m \times n}$ (similarly, columns of each $Y(i) \in \mathbb{R}^{n \times p}$), $i \in [-T + 1 : J + T]$, be divided into $x$ subchunks (similarly, $y$ subchunks) as in Section 3.1. Again, we set $X(l') \triangleq \mathbf{0}_{m \times n}, Y(l') \triangleq \mathbf{0}_{n \times p}$, whenever $l' \notin [1 : J]$, as only jobs 1 to $J$ are non-trivial. Let $k_N \triangleq PW - N$, $n_N \triangleq W$. Recall the definition of polynomials $\mathcal{X}_i(\Theta) \triangleq \sum_{i'=1}^{x} X(i; i') \Theta^{y(i'-1)}$, $\mathcal{Y}_i(\Theta) \triangleq \sum_{j'=1}^{y} Y(i; j') \Theta^{j'-1}$, whose coefficients are subchunks of $X(i)$ and $Y(i)$, respectively. Here $x, y$ are chosen so that $xy = k_N$. The scheme operates with a delay $T = n_N - 1$.

For each $i \in [-T + 1 : J + T]$, the master keeps an ordered list $\mathcal{U}_i$ of the form:

$$\mathcal{U}_i \triangleq \{(X(i; 1), Y(i; 1)), (X(i; 1), Y(i; 2)), \ldots, (X(i; x), Y(i; y - 1)), (X(i; x), Y(i; y))\},$$

which consists of $xy$ entries. Each entry in the ordered list $\mathcal{U}_i$ is a 2-tuple composed of two matrices; a subchunk of $X(i)$ and $Y(i)$ each. The ordering of these entries within $\mathcal{U}_i$ is in such a way that multiplication of the two subchunks in the $n'$-th entry, for $1 \leq n' \leq xy$, gives the coefficient corresponding to $\Theta^{n'-1}$ of the polynomial $\mathcal{X}_i(\Theta) * \mathcal{Y}_i(\Theta)$.

In each round-$t$ ($t \in [1 : J + T]$), the master assigns $\ell = n_N = T + 1$ mini-tasks to each worker (some of these mini-tasks can be trivial mini-tasks). For $l \in [1 : \ell]$, the $l$-th mini-task assigned to worker-$j$ ($j \in [1 : P]$) in round-$t$ corresponds to job-$(t - l + 1)$ in the following sense. The input matrix-pair $(\tilde{X}_j(t; l), \tilde{Y}_j(t; l))$ for this mini-task will be either (i) *uncoded*: one of the entries of $\mathcal{U}_{t-l+1}$ or else (ii) *coded*: evaluation of $(\mathcal{X}_{t-l+1}(\Theta), \mathcal{Y}_{t-l+1}(\Theta))$ at some $\theta_{j,t,l} \in \mathbb{R}$. Each worker-$j$ processes the $\ell$ mini-tasks one after another and communicates each result to the master. As can be seen from Fig. 6, mini-tasks corresponding to each job-$i$ assigned to any worker-$j$ are distributed in a diagonal manner across rounds.

In Algorithm 2, we describe how master selects the input matrix-pair $(\tilde{X}_j(t; l), \tilde{Y}_j(t; l))$ for the $l$-th mini-task to be assigned to each worker-$j$ in round-$t$. We assume that job-$i'$ is finished, whenever $i' \notin [1 : J]$. Furthermore, when $l = 1$ in the outermost loop of Algorithm 2, $\mathcal{J}$ in line-5 becomes the empty set, as mini-tasks are indexed in the range $[1 : \ell]$.

Worker-$j$

| | | | | | | |
|---|---|---|---|---|---|---|
| ...... | 2 | 3 | 4 | 5 | 6 | ...... |
| | 1 | 2 | 3 | 4 | 5 | |
| | 0 | 1 | 2 | 3 | 4 | $n_N = T + 1 = 5$ |
| | −1 | 0 | 1 | 2 | 3 | |
| | −2 | −1 | 0 | 1 | 2 | |
| | 2 | 3 | 4 | 5 | 6 | |

Round $\longrightarrow$

Figure 6: Each rectangle here depicts a mini-task. The number inside each rectangle is indicative of the job corresponding to that particular mini-task. Uncolored rectangles indicate trivial mini-tasks, which correspond to trivial jobs.

We claim that all the uncoded mini-tasks corresponding to each job-$i$ will succeed in one of the rounds $[i : i+T]$ (as per the straggler model). This claim can be verified as follows. For $l \in [1 : T+1]$, recall that mini-tasks corresponding to job-$(t - l + 1)$ appear as the first mini-task of every worker in round-$(t - l + 1)$, second mini-task in round-$(t - l + 2)$, . . ., $(T + 1)$-th mini-task in round-$(t - l + 1 + T)$. The master makes the $l$-th mini-task of worker-$j$ in round-$t$ (corresponding to job-$(t - l + 1)$) uncoded only if worker-$j$ cannot be a straggler in at least one of the rounds $[t : t - l + 1 + T]$ as per the straggler model (see line-13 of Algorithm 2). Any such uncoded mini-task will be successfully processed by worker-$j$ in the first round among $[t : t - l + 1 + T]$ where worker-$j$ is not a straggler, as a failed mini-task of worker-$j$ in a round is reattempted in the next round (see line-7 of the algorithm).

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

---

divide the rows of $X(i)$ into $x = 4$ subchunks and columns of $Y(i)$ into $y = 2$ subchunks:

$$X(i) \triangleq \begin{bmatrix} X(i;1) \\ X(i;2) \\ X(i;3) \\ X(i;4) \end{bmatrix}, \quad Y(i) \triangleq [\ Y(i;1) \quad Y(i;2)\ ],$$

where $X(i; j') \in \mathbb{R}^{\frac{m}{4} \times n}$, $Y(i; l') \in \mathbb{R}^{n \times \frac{p}{2}}$ for $j' \in [1 : 4], l' \in [1 : 2]$. Let $t \in [1 : J + T]$. In round-$t$, each worker will be assigned a task consisting of $\ell = 4$ mini-tasks. Each mini-task involves multiplication of two matrices having dimensions $\frac{m}{4} \times n$ and $n \times \frac{p}{2}$, respectively. There are $T + 1$ such mini-tasks assigned to each worker, in each round. Hence, the task load is $4 * \frac{mnp}{8} = \frac{mnp}{2}$. Thus, normalized task load equals $\frac{1}{2}$. The scheme discussed here operates with a delay $T = 3$. Let $\mathcal{X}_i(\Theta) \triangleq X(i;1) + X(i;2)\Theta^2 + X(i;3)\Theta^4 + X(i;4)\Theta^6$, $\mathcal{Y}_i(\Theta) \triangleq Y(i;1) + Y(i;2)\Theta$.

We have:

$$\mathcal{U}_i \triangleq \{(X(i;1), Y(i;1)), (X(i;1), Y(i;2)), (X(i;2), Y(i;1)), \ldots, (X(i;4), Y(i;2))\},$$

For $l \in [1 : 4]$, $j \in [1 : 3]$, $t \in [1 : J + T]$, the $l$-th mini-task assigned to worker-$j$ in round-$t$ (corresponding to job-$(t - l + 1)$) is non-trivial iff both following conditions are true (i) $(t - l + 1) \in [1 : J]$ and (ii) master cannot decode job-$(t - l + 1)$ with the available mini-task results from rounds $[t - l + 1 : t - 1]$. A non-trivial mini-task can either be coded or uncoded. If the mini-task is uncoded, the input matrices $(\tilde{X}_j(t; l), \tilde{Y}_j(t; l))$ for the mini-task will be an entry of $\mathcal{U}_{t-l+1}$. If the mini-task is coded, the master set $(\tilde{X}_j(t; l), \tilde{Y}_j(t; l)) = (X'(t - l + 1; i'), Y'(t - l + 1; i'))$, where $X'(t - l + 1; i'), Y'(t - l + 1; i')$ are evaluations of $\mathcal{X}_{t-l+1}(\Theta)$ and $\mathcal{Y}_{t-l+1}(\Theta)$, respectively, at some $\theta_{i'} \in \mathbb{R}$.

Consider the assignment of mini-tasks corresponding to job-3 in Fig. 7. These mini-tasks will be distributed across workers from round-3 till round-6 (recall that delay $T = 3$ in this example).

*Round*-3: In rounds 1 and 2, the master has seen three straggling rounds faced in total by all the three workers. Hence, none of the workers can be a straggler in both rounds 3 and 4, as it violates the straggler model assumption. Thus, any mini-task corresponding to job-3 assigned to a worker-$j$ in round-3 will succeed in one of the rounds $\{3, 4\}$ (because of the possibility of reattempts). Exploiting this scenario, master pushes uncoded mini-tasks corresponding to job-3 to all the workers in round-3 (as the first mini-task of every worker).

*Round*-4: As worker-2 is a straggler in round-3, the mini-task corresponding to job-3 assigned in round-3 will be reattempted as the second mini-task of worker-2 in round-4. The master has seen three straggling rounds faced by workers in total, in rounds 2 and 3. Hence, it is not possible for any worker-$j$ to be a straggler in rounds 4 and 5 both. Thus for workers 1 and 3, uncoded mini-tasks corresponding to job-3 will be assigned by the master.

*Round*-5: Master now has history of all the straggling rounds till round-4. It is possible that a given worker-$j$ can be a straggler in both rounds 5 and 6. Thus, master assigns coded mini-tasks corresponding to job-3 to all the workers.

*Round*-6: As mini-task assigned to worker-3 failed in round-5, it will be reattempted in round-6. As workers 1 and 2 can possibly be a straggler in round-6, coded mini-tasks corresponding to job-3 will be assigned to workers 1 and 2.

Once round-6 is over, master has access to results of 8 mini-tasks corresponding to job-3; five of them uncoded and remaining three are coded. The results of uncoded mini-tasks available to the master are: $\{X(3;1)*Y(3;1), X(3;1)*Y(3;2), X(3;2)*Y(3;1), X(3;2)*Y(3;2), X(3;3)*Y(3;1)\}$. The results of coded mini-tasks available are $\{X'(3;1)*Y'(3;1), X'(3;2)*Y'(3;2), X'(3;5)*Y'(3;5)\}$. The coded mini-task results correspond to evaluations of $\mathcal{X}_3(\Theta) * \mathcal{Y}_3(\Theta)$ at three distinct non-zero points $\theta_1, \theta_2, \theta_5$. As the results of uncoded mini-tasks are nothing but coefficients of $\{\Theta^0, \Theta, \ldots, \Theta^4\}$, the master effectively has three evaluations of the polynomial $X(3;3)*Y(3;2)+X(3;4)*Y(3;1)\Theta + X(3;4)*Y(3;2)\Theta^2$ at $\theta_1, \theta_2, \theta_5$. The coefficients $X(3;3)*Y(3;2)$, $X(3;3)*Y(3;2)$ and $X(3;3)*Y(3;2)$ can thus be obtained via polynomial interpolation. Finally, $X(3)*Y(3)$ is obtained by simply arranging all the coefficients of $\mathcal{X}_3(\Theta) * \mathcal{Y}_3(\Theta)$ in the form of an $m \times p$ matrix.

**Remark 3.3** (Extension of the IDIP scheme for the bursty straggler model). IDIP scheme for the $(B, \epsilon, W)$-bursty straggler model follows exactly the same manner as in the case of arbitrary straggler model. Let $B < W$. Instead of $k_N$, $n_N$, we use here the parameters $k_B \triangleq P(W - 1 + B) - B\epsilon$, $n_B \triangleq W - 1 + B$, respectively. Delay parameter, $T \triangleq W - 2 + B = n_B - 1$. When $B = W$, the straggler model degenerates to a scenario where there can be $\epsilon$ stragglers in each round. In this case, we can deploy polynomial code [3], which is resilient against $S = \epsilon$ stragglers in each round.

### 3.3 Optimality of DIP, IDIP coding schemes under arbitrary, bursty straggler model assumptions

The following theorems provide lower bounds for worst-case task load of any $z = 1$ scheme resilient against straggler patterns conforming to arbitrary or bursty straggler models in all the rounds $[1 : J + T]$, under the system model discussed in Section 2.1. Proofs are deferred to Appendices A and B.

**Theorem 3.1** (Worst-case load for arbitrary straggler model). *Let $J \to \infty$ and $T < \infty$. The worst-case normalized task load $L$ under the $(N, W)$-arbitrary straggler model is lower bounded as:*

$$L \geq L^* = \frac{1}{P - \frac{N}{W}}. \tag{2}$$

**Theorem 3.2** (Worst-case load for bursty straggler model). *Let $J \to \infty$ and $T < \infty$. The worst-case normalized task load $L$ under the $(B, \epsilon, W)$-bursty straggler model is lower bounded as:*

$$L \geq L^* = \begin{cases} \frac{W-1+B}{P(W-1+B)-B\epsilon}, & \text{if } B < W, \\ \frac{1}{P-\epsilon}, & \text{if } B = W. \end{cases} \tag{3}$$

#### 3.3.1 Optimality of DIP scheme

DIP coding scheme does not require any underlying assumptions on the straggler model. In the following, we state results showing that if the straggler pattern conforms to arbitrary/bursty straggler model, the scheme provides optimal worst-case normalized task load (under a certain choice of parameters).

| Worker | Round 1 | Round 2 | Round 3 | Round 4 | Round 5 | Round 6 | Round 7 | Round 8 |
|---|---|---|---|---|---|---|---|---|
| Worker 1 | $X'(1;1)$, $Y'(1;1)$ | $X'(2;1)$, $Y'(2;1)$ | $X(3;1)$, $Y(3;1)$ | $X(4;1)$, $Y(4;1)$ | $X'(5;1)$, $Y'(5;1)$ | $X'(6;1)$, $Y'(6;1)$ | $X(7;1)$, $Y(7;1)$ | $X(8;1)$, $Y(8;1)$ |
| | --- | $X'(1;1)$, $Y'(1;1)$ | $X(2;1)$, $Y(2;1)$ | $X(3;2)$, $Y(3;2)$ | $X'(4;1)$, $Y'(4;1)$ | $X'(5;4)$, $Y'(5;4)$ | $X'(6;1)$, $Y'(6;1)$ | $X(7;1)$, $Y(7;1)$ |
| | --- | --- | $X(1;1)$, $Y(1;1)$ | $X(2;1)$, $Y(2;2)$ | $X'(3;1)$, $Y'(3;1)$ | $X'(4;4)$, $Y'(4;4)$ | $X'(5;4)$, $Y'(5;4)$ | $X'(6;1)$, $Y'(6;1)$ |
| | --- | --- | --- | $X(1;1)$, $Y(1;2)$ | $X'(2;4)$, $Y'(2;4)$ | $X'(3;4)$, $Y'(3;4)$ | $X'(4;4)$, $Y'(4;4)$ | $X'(5;4)$, $Y'(5;4)$ |
| Worker 2 | $X'(1;2)$, $Y'(1;2)$ | $X'(2;2)$, $Y'(2;2)$ | $X(3;1)$, $Y(3;2)$ | $X(4;1)$, $Y(4;2)$ | $X'(5;2)$, $Y'(5;2)$ | $X'(6;2)$, $Y'(6;2)$ | $X(7;1)$, $Y(7;2)$ | $X(8;1)$, $Y(8;2)$ |
| | --- | $X'(1;4)$, $Y'(1;4)$ | $X'(2;2)$, $Y'(2;2)$ | $X(3;1)$, $Y(3;2)$ | $X'(4;2)$, $Y'(4;2)$ | $X'(5;5)$, $Y'(5;5)$ | $X(6;1)$, $Y(6;1)$ | $X(7;2)$, $Y(7;2)$ |
| | --- | --- | $X'(1;4)$, $Y'(1;4)$ | $X'(2;2)$, $Y'(2;2)$ | $X'(3;2)$, $Y'(3;2)$ | $X'(4;5)$, $Y'(4;5)$ | $X(5;1)$, $Y(5;1)$ | $X(6;1)$, $Y(6;2)$ |
| | --- | --- | --- | $X'(1;4)$, $Y'(1;4)$ | $X'(2;5)$, $Y'(2;5)$ | $X'(3;5)$, $Y'(3;5)$ | $X'(4;6)$, $Y'(4;6)$ | $X(5;1)$, $Y(5;2)$ |
| Worker 3 | $X'(1;3)$, $Y'(1;3)$ | $X'(2;3)$, $Y'(2;3)$ | $X(3;2)$, $Y(3;1)$ | $X(4;2)$, $Y(4;1)$ | $X'(5;3)$, $Y'(5;3)$ | $X'(6;3)$, $Y'(6;3)$ | $X(7;2)$, $Y(7;1)$ | $X(8;2)$, $Y(8;1)$ |
| | --- | $X'(1;5)$, $Y'(1;5)$ | $X'(2;3)$, $Y'(2;3)$ | $X(3;3)$, $Y(3;1)$ | $X'(4;3)$, $Y'(4;3)$ | $X'(5;3)$, $Y'(5;3)$ | $X'(6;3)$, $Y'(6;3)$ | $X(7;3)$, $Y(7;1)$ |
| | --- | --- | $X'(1;5)$, $Y'(1;5)$ | $X(2;2)$, $Y(2;1)$ | $X'(3;3)$, $Y'(3;3)$ | $X'(4;3)$, $Y'(4;3)$ | $X'(5;3)$, $Y'(5;3)$ | $X(6;2)$, $Y(6;1)$ |
| | --- | --- | --- | $X(1;2)$, $Y(1;1)$ | $X'(2;6)$, $Y'(2;6)$ | $X'(3;3)$, $Y'(3;3)$ | $X'(4;3)$, $Y'(4;3)$ | $X(5;2)$, $Y(5;1)$ |

Round $\longrightarrow$ 1 2 3 4 5 6 7 8

Figure 7: An illustration of the task assignment to workers when the straggler pattern conforms to the $(N = 4, W = 4)$-arbitrary straggler model. Each rectangular box here corresponds to a mini-task involving multiplication of two matrices having sizes $\frac{m}{4} \times n$ and $n \times \frac{p}{2}$, respectively. Stragglers are identified using shaded boxes. Uncoded and coded mini-tasks are indicated in green and red, respectively.

**Optimality under arbitrary straggler model** Let parameters $x, y, \lambda, T$ of the DIP coding scheme be such that $xy = PW - N, \lambda = P, T = W - 1$. It can be shown that the worst-case normalized task load matches (2).

We outline the proof as follows. The number of mini-tasks assigned per worker in round-$t$ is given by $\ell_t = T + \max(\lceil \frac{xy - \gamma_t}{\lambda} \rceil, 0)$, where $\gamma_t$ is the number of results of mini-tasks corresponding to job-$(t - T)$ received by master from all the previous rounds prior to round-$t$. The first mini-task corresponding to job-$(t - T)$ gets assigned in round-$(t - T)$. As per the straggler model, in any sliding window of $W = T + 1$ rounds, the workers face at most $N$ straggling rounds in total. For the $T = W - 1$ rounds in the range $[t - T : t - 1]$, let there be $N - \delta_1$ straggling rounds in total, for some $\delta_1 \geq 0$. Hence, from the $P(W - 1)$ mini-tasks corresponding to job-$(t - T)$ assigned in rounds $[t - T : t - 1]$, the master must have received results of $P(W - 1) - N + \delta_1$ mini-tasks prior to round-$t$. Thus $\gamma_t = P(W - 1) - N + \delta_1$ and we have $\ell_t = T + \max(\lceil \frac{xy - \gamma_t}{\lambda} \rceil, 0) = T + \max(\lceil \frac{P - \delta_1}{P} \rceil, 0) \leq T + 1 = W$. If $\gamma_t \geq xy$, the master already has enough results of mini-tasks corresponding to job-$(t - T)$, hence no mini-task corresponding to job-$(t - T)$ has to be assigned in round-$t$ and $\ell_t$ will be equal to $T$. If $\gamma_t < xy$, $\ell_t$ will be equal to $T + 1$ and each worker will be having one mini-task corresponding to job-$(t - T)$. As per the straggler model, in round-$t$, there can be at most $\delta_1$ stragglers. Hence, the master will receive at least $P - \delta_1$ more mini-task results corresponding to job-$(t - T)$ in round-$t$. Thus, in total, master will receive at least $P(W - 1) - N + \delta_1 + P - \delta_1 = xy$ mini-tasks corresponding to job-$(t - T)$ from all the rounds $[t - T : t]$ and hence, job-$(t - T)$ can be successfully finished after round-$t$. As $\ell_t \leq W$, we have a normalized task load of at most $\frac{W}{xy} = \frac{W}{PW - N}$, which matches (2).

**Optimality under bursty straggler model** Let $B < W$ and parameters $x, y, \lambda, T$ of the DIP coding scheme be such that $xy = P(W - 1 + B) - B\epsilon, \lambda = P, T = W - 2 + B$. In an analogous manner, it can be argued that the normalized task load matches (3).

### 3.3.2 Optimality of IDIP scheme

Consider the arbitrary straggler model and the associated IDIP scheme presented in Section 3.2.3. Clearly, by design, IDIP scheme is resilient against any straggler pattern conforming to the $(N, W)$-arbitrary straggler model. The number of mini-tasks assigned to each worker is $n_N$ and normalized task load $\frac{n_N}{k_N}$, which matches (2). The optimality of IDIP scheme for bursty straggler model (see Remark 3.3) can also be shown in a similar manner.

## 4 Numerical results

In this section, we analytically compare the performance of our two schemes with the polynomial coding scheme. We consider in this section, a probabilistic, i.i.d. straggler model, i.e., any worker in any given round will be a straggler with probability $\delta$. We make the simplifying assumption that for a normalized task load $L(t)$ in round-$t$, a non-straggler takes $\tau L(t)$ seconds to finish the task. Conversely, a straggler takes $\alpha \tau L(t)$ seconds to finish the task ($\alpha > 1$). As randomness in this system is occurring only due to the straggler model, we set $\mu = 0$ (the parameter used by master to detect stragglers in each round). Let $R_J$ denote the time required to complete $J$ jobs. We have $\hat{R} \triangleq \lim_{J \to \infty} \frac{\mathbb{E}[R_J]}{J\tau}$.

### 4.1 Uncoded scheme

In this scheme, no coding is performed while generating input matrix-pairs for mini-tasks. For $t \in [1 : J]$, let rows of each $X(t) \in \mathbb{R}^{m \times n}$ (similarly, columns of each $Y(t) \in \mathbb{R}^{n \times p}$) be divided into $x$ subchunks $\{X(t; 1), X(t; 2), \dots, X(t; x)\}$ (similarly, $y$ subchunks $\{Y(t; 1), Y(t; 2), \dots, Y(t; y)\}$), where $xy = P$, the total number of workers. In the beginning of round-$t$, master will communicate the subchunks $\{X(t; i), Y(t; j)\}$ to worker-$((i - 1)y + j)$, where $1 \leq i \leq x$, $1 \leq j \leq y$. Each worker-$((i - 1)y + j)$ computes $X(t; i) * Y(t; j)$ and returns this matrix-product to the master. The master can compute $X(t) * Y(t)$ when it receives results from all workers. Here, normalized task load $L(t) = \frac{1}{xy} = \frac{1}{P}$.

**Proposition 1.** We have:

$$\hat{R}^{\text{uncoded}}_{\alpha, P, \delta} = \frac{\alpha p^{\text{uncoded}}_{P,\delta} + (1 - p^{\text{uncoded}}_{P,\delta})}{P},$$

where $p^{\text{uncoded}}_{P,\delta} = 1 - (1 - \delta)^P$.

*Proof.* Note that $p^{\text{uncoded}}_{P,\delta}$ is the probability that at least one of the workers is a straggler in any given round-$t$, $t \in [1 : J]$. As master requires results from all the workers in round-$t$ to complete job-$t$, the duration $r_t$ of round-$t$ is distributed as:

$$r_t = \begin{cases} \frac{\tau}{P}, & w.p. \ 1 - p^{\text{uncoded}}_{P,\delta} \\ \frac{\alpha\tau}{P}, & w.p. \ p^{\text{uncoded}}_{P,\delta} \end{cases}.$$

Thus, we have:

$$\mathbb{E}[R_J] = \mathbb{E}[\sum_{t=1}^{J} r_t] = J\mathbb{E}[r_t] = J\tau \frac{\alpha p^{\text{uncoded}}_{P,\delta} + (1 - p^{\text{uncoded}}_{P,\delta})}{P},$$

and the proof follows. $\qquad\square$

### 4.2 Polynomial coding scheme

For $t \in [1 : J]$, consider the polynomials $\mathcal{X}_t(\Theta) \triangleq \sum_{i'=1}^{x} X(t; i')\Theta^{y(i'-1)}$, $\mathcal{Y}_t(\Theta) \triangleq \sum_{j'=1}^{y} Y(t; j')\Theta^{j'-1}$. In the beginning of round-$t$, master transmits $(\mathcal{X}_t(\Theta)|_{\Theta=\theta_{j,t}}, \mathcal{Y}_t(\Theta)|_{\Theta=\theta_{j,t}})$ to each worker-$j$, $j \in [1 : P]$. Here, $\{\theta_{j,t}\}_{j=1}^{P}$ are $P$ distinct evaluation points drawn from $\mathbb{R}$. Worker-$j$ attempts to compute $\mathcal{X}_t(\Theta)|_{\Theta=\theta_{j,t}} * \mathcal{Y}_t(\Theta)|_{\Theta=\theta_{j,t}}$. Owing to the algebraic properties of $\mathcal{X}_t(\Theta)$ and $\mathcal{Y}_t(\Theta)$, master can compute $X(t) * Y(t)$ via a decoding step, if it receives results from any $xy < P$ workers. Hence, the scheme can tolerate $S \triangleq (P - xy)$ stragglers. As $\mathcal{X}_t(\Theta)|_{\Theta=\theta_{j,t}} \in \mathbb{R}^{\frac{m}{x} \times n}$ and $\mathcal{Y}_t(\Theta)|_{\Theta=\theta_{j,t}} \in \mathbb{R}^{n \times \frac{p}{y}}$, normalized task load $L(t) = \frac{1}{xy} = \frac{1}{P-S}$.

**Proposition 2.** We have:
$$\hat{R}^{\text{poly}}_{\alpha,S,P,\delta} = \frac{\alpha p^{\text{poly}}_{S,P,\delta} + (1 - p^{\text{poly}}_{S,P,\delta})}{P - S},$$
where $p^{\text{poly}}_{S,P,\delta} \triangleq \sum_{i=S+1}^{P} \binom{P}{i}\delta^i(1-\delta)^{P-i}$.

*Proof.* We begin with noting that $p^{\text{poly}}_{S,P,\delta}$ is the probability that there are at least $S+1$ stragglers in any given round-$t$, $t \in [1 : J]$. The master requires results from $P - S$ workers to compute the matrix product $X(t) * Y(t)$. Thus, the duration $r_t$ of round-$t$ is distributed as:

$$r_t = \begin{cases} \frac{\tau}{P-S}, & w.p. \ \ 1 - p^{\text{poly}}_{S,P,\delta} \\ \frac{\alpha\tau}{P-S}, & w.p. \ \ p^{\text{poly}}_{S,P,\delta} \end{cases}.$$

Hence:

$$\mathbb{E}[R_J] = \mathbb{E}[\sum_{t=1}^{J} r_t] = J\mathbb{E}[r_t] = J\tau \frac{\alpha p^{\text{poly}}_{S,P,\delta} + (1 - p^{\text{poly}}_{S,P,\delta})}{P - S},$$

and the proof follows. ☐

### 4.3 IDIP coding scheme

Here, we study the performance of the IDIP coding scheme designed for the $(N, W)$-arbitrary straggler model, although the stragglers are sampled from an i.i.d. model. Recall that the IDIP scheme is resilient against upto $N$ stragglers in any sliding window consisting of $W$ consecutive rounds.

**Proposition 3.** We have:

$$\hat{R}^{\text{IDIP}}_{\alpha,N,W,P,\delta} \leq \frac{\alpha p^{\text{IDIP}}_{N,W,P,\delta} + (1 - p^{\text{IDIP}}_{N,W,P,\delta})}{P - \frac{N}{W}},$$

where $p^{\text{IDIP}}_{N,W,P,\delta} \triangleq \sum_{i=N+1}^{PW} \binom{PW}{i}\delta^i(1-\delta)^{PW-i}$.

*Proof.* Let $t \in [1 : J + T], j \in [1 : P]$. Recall the waiting strategy employed by master (Section 3.2.3), if the actually experienced straggler pattern does not conform to the $(N, W)$-arbitrary straggler model in any round-$t$. As we assume no randomness in processing times across workers, the waiting strategy simplifies as follows. After $\tau L(t)$ seconds into round-$t$, if straggler pattern from rounds $[1 : t]$ does not conform to the $(N, W)$-arbitrary straggler model, the master will wait for all the remaining workers to return their mini-task results (i.e., round-$t$ will have duration $r_t = \alpha\tau L(t)$). Let $M_j(t)$ be an indicator function taking value 1 if and only if worker-$j$ is a straggler after $\tau L(t)$ seconds into round-$t$. Because of the i.i.d. straggler model, $M_j(t) = 1$ with probability $\delta$ (independently for each $j, t$). Note that master will be forced to wait in round-$t$ *only if* $\sum_{i=t-T}^{t} \sum_{j=1}^{P} M_j(i) > N$. Let $p_{\text{wait}}(t)$ denote the probability that master will wait in round-$t$. We have:

$$
\begin{aligned}
p_{\text{wait}}(t) &\leq \Pr\left\{ \sum_{i=t-T}^{t} \sum_{j=1}^{P} M_j(i) > N \right\} \\
&= \sum_{i=N+1}^{PW} \binom{PW}{i}\delta^i(1-\delta)^{PW-i} \\
&\triangleq p^{\text{IDIP}}_{N,W,P,\delta}.
\end{aligned}
\tag{4}
$$

Clearly, the duration $r_t$ of round-$t$ satisfies:

$$r_t = \begin{cases} \tau L(t), & w.p. \ \ 1 - p_{\text{wait}}(t) \\ \alpha\tau L(t), & w.p. \ \ p_{\text{wait}}(t) \end{cases} \tag{5}$$

From the design of the IDIP scheme, we have:

$$L(t) \leq \frac{W}{WP - N}. \tag{6}$$

We have an inequality in (6), as some of the mini-tasks (among the total $W$ mini-tasks per worker in each round-$t$ of the IDIP scheme) might be trivial mini-tasks. Trivial mini-tasks do not contribute towards processing time and computational load. Hence, the RHS of (6) is a worst-case estimate of normalized task load. We now have:

$$
\begin{aligned}
\mathbb{E}[R_J] &= \mathbb{E}[\sum_{t=1}^{J+T} r_t] \\
&\overset{(a)}{=} \mathbb{E}[\sum_{t=1}^{J+T} \tau \frac{(1 - p_{\text{wait}}(t)) + \alpha p_{\text{wait}}(t)}{\beta} L(t)] \\
&\overset{(b)}{\leq} (J+T)\tau \frac{\alpha W p_{N,W,P,\delta}^{\text{IDIP}} + W(1 - p_{N,W,P,\delta}^{\text{IDIP}})}{WP - N},
\end{aligned}
$$

where $(a)$ follows from (5), $(b)$ follows from (4) and (6). Hence, $\hat{R}_{\alpha,N,W,P,\delta}^{\text{IDIP}} = \lim_{J \to \infty} \frac{\mathbb{E}[R_J]}{J\tau} \leq \frac{\alpha p_{N,W,P,\delta}^{\text{IDIP}} + (1 - p_{N,W,P,\delta}^{\text{IDIP}})}{P - \frac{N}{W}}$. $\qquad \square$

In order to compare $\hat{R}_{\alpha,N,W,P,\delta}^{\text{IDIP}}$ with $\hat{R}_{\alpha,S,P,\delta}^{\text{poly}}$, set $N = SW$. The probability that there are more than $N$ straggling rounds across $P$ workers in $W$ consecutive rounds is given by $p_{N,W,P,\delta}^{\text{IDIP}}$. By pigeonhole principle, at least one round will now have more than $S$ stragglers. The probability that a given round has more than $S$ stragglers is given by $p_{S,P,\delta}^{\text{poly}}$. Thus $p_{N,W,P,\delta}^{\text{IDIP}} \leq p_{S,P,\delta}^{\text{poly}}$ and hence, $\hat{R}_{\alpha,N,W,P,\delta}^{\text{IDIP}} \leq \hat{R}_{\alpha,S,P,\delta}^{\text{poly}}$. In this case, the worst-case normalized task load for both the schemes is given by $\frac{W}{WP - N} = \frac{W}{WP - SW} = \frac{1}{P - S}$.

## 4.4 DIP coding scheme

In the analysis of DIP scheme, we consider two regimes; (i) infinite $T$ and (ii) finite $T$.

### 4.4.1 Unconstrained delay ($T = \infty$)

Let $\beta = xy$, which is the total number of mini-tasks needed for completing each job. When $T = \infty$, the DIP scheme described in Section 3.1 simplifies to the following. For $t \in [1 : \infty]$ and $l \in [1 : \infty]$, the $l$-th mini-task of any worker-$j$ ($j \in [1 : P]$) in round-$t$ corresponds to job-$(t - l + 1)$. As only jobs in the range $[1 : J]$ are non-trivial, even though the range of $l$ varies over $[1 : \infty]$, only a finite number of mini-tasks will be non-trivial. The parameter $\lambda$ is inactive when $T = \infty$ (we simply set $\lambda \triangleq \infty$ to emphasize that it is inactive). Clearly, decoding of job-$i$ ($i \in [1 : J]$) happens when master collects $\beta$ mini-task results corresponding to job-$i$. The expected number of rounds required to finish processing a job (denoted by $f_{\beta,P,\delta}$), as we will see, is a finite quantity (also, see Fig. 8). Thus, despite setting $T = \infty$, each job will be finished in finite number of rounds.

Let $C(t)$ denote the number of workers who return results of all the mini-tasks assigned to them in round-$t$. Let the number of non-stragglers in round-$t$ be $\hat{S}(t)$. If $\hat{S}(t) \neq 0$, we have $C(t) = \hat{S}(t)$. However, if $\hat{S}(t) = 0$, i.e., if all the workers in round-$t$ are stragglers, all of them will be incurring $\alpha\tau L(t)$ seconds to finish the mini-tasks assigned to them. As master identifies stragglers by comparing the processing times of workers with respect to the fastest worker, master will receive mini-task results from every worker. Thus, when $\hat{S}(t) = 0$, we have $C(t) = P$. Clearly, $C(t)$ is distributed as:

$$
\mathbb{P}[C(t) = k] \triangleq q_k = \begin{cases} \binom{P}{k}(1-\delta)^k \delta^{P-k}, & \text{if } 1 \leq k \leq P-1 \\ \binom{P}{0}\delta^P + \binom{P}{P}(1-\delta)^P, & \text{if } k = P \end{cases}, \tag{7}
$$

where $\mathbb{P}[E]$ denotes probability of some event $E$. For $i \in [1 : J]$, let $t_i$ denote the number of rounds taken to complete job-$i$. Since, $C(t)$ workers return all their mini-task results in round-$t$, $t_i$ is the

Figure 8: Let $P = 4$. We illustrate here how $f_{\beta,\delta,P}$ varies for different values of $\beta$ and $\delta$.

smallest $t_i^*$ satisfying:

$$\sum_{t=i}^{i+t_i^*-1} C(t) \geq \beta.$$

We are slightly conservative here as even though stragglers cannot provide all the mini-task results, it can still provide some mini-task results, which can perhaps contribute towards making $t_i$ smaller. Let $f_{\beta,\delta,P} \triangleq \mathbb{E}[t_i]$. It can be easily verified that $f_{1,\delta,P} = 1$. We will now recursively compute the value of $f_{\beta,\delta,P}$. First, let us assume that $\beta \leq P$. If $C(i) \geq \beta$, we will have $t_i = 1$. Thus, we have:

$$f_{\beta,\delta,P} = \sum_{k=1}^{\beta-1} q_k(1 + f_{\beta-k,\delta,P}) + \sum_{k=\beta}^{P} q_k = 1 + \sum_{k=1}^{\beta-1} q_k f_{\beta-k,\delta,P}. \tag{8}$$

For the case $\beta > P$, job-$i$ clearly cannot be finished in one round and $t_i \geq 2$. Here, we have:

$$f_{\beta,\delta,P} = \sum_{k=1}^{P} q_k(1 + f_{\beta-k,\delta,P}) = 1 + \sum_{k=1}^{P} q_k f_{\beta-k,\delta,P}. \tag{9}$$

Combining (8) and (9),

$$f_{\beta,\delta,P} = \begin{cases} 1 + \sum_{k=1}^{\beta-1} q_k f_{\beta-k,\delta,P}, & \text{if } \beta \leq P, \\ 1 + \sum_{k=1}^{P} q_k f_{\beta-k,\delta,P}, & \text{if } \beta > P. \end{cases} \tag{10}$$

Let $\ell_t$ denote the number of non-trivial mini-tasks assigned per worker in round-$t$. Hence, the normalized task load (considering non-trivial mini-tasks alone) per worker in round-$t$, $L(t) = \frac{\ell_t}{\beta}$. In the case of $T = \infty$, the only way round-$t$ will have duration $\frac{\alpha \ell_t \tau}{\beta}$ is when all the workers in round-$t$ are stragglers. Let $p_{\delta,P}^{\text{all}} \triangleq \delta^P$ be the probability that all the workers in a given round-$t$ are stragglers. The duration $r_t$ of round-$t$ satisfies:

$$r_t = \begin{cases} \frac{\ell_t \tau}{\beta}, & w.p. \quad 1 - p_{\delta,P}^{\text{all}}, \\ \frac{\alpha \ell_t \tau}{\beta}, & w.p. \quad p_{\delta,P}^{\text{all}}. \end{cases} \tag{11}$$

As each non-trivial mini-task corresponds to some job-$i$, based on double counting, we have:

$$\sum_{t=1}^{\infty} \ell_t = \sum_{i=1}^{J} t_i. \tag{12}$$

Now, we can calculate the average run-time for finishing $J$ jobs as:

$$\begin{aligned} \mathbb{E}[R_J] &= \mathbb{E}[\sum_{t=1}^{\infty} r_t] = \mathbb{E}[\tau \frac{(1 - p_{\delta,P}^{\text{all}}) + \alpha p_{\delta,P}^{\text{all}}}{\beta} \sum_{t=1}^{\infty} \ell_t] \\ &= \tau \frac{(1 - p_{\delta,P}^{\text{all}}) + \alpha p_{\delta,P}^{\text{all}}}{\beta} \sum_{i=1}^{J} \mathbb{E}[t_i] \\ &= \frac{J \tau f_{\beta,\delta,P}}{\beta}((1 - p_{\delta,P}^{\text{all}}) + \alpha p_{\delta,P}^{\text{all}}). \end{aligned} \tag{13}$$

It follows that:

$$\hat{R}^{\text{DIP}}_{\alpha,\beta,T=\infty,\lambda=\infty,P,\delta} = \frac{f_{\beta,P,\delta}}{\beta}(\alpha p^{\text{all}}_{\delta,P} + (1 - p^{\text{all}}_{\delta,P})).$$

The expected number of workers $\mathbb{E}[C(t)]$ returning their mini-tasks in round-$t$ is given by $P(1 - \delta) + P\delta^P$. Hence, $f_{\beta,P,\delta}$ can be approximated as $\frac{\beta}{P(1-\delta)+P\delta^P}$. Thus, $\hat{R}^{\text{DIP}}_{\alpha,\beta,T=\infty,\lambda=\infty,P,\delta} \approx \frac{\alpha p^{\text{all}}_{\delta,P}+(1-p^{\text{all}}_{\delta,P})}{P(1-\delta+\delta^P)}$. Note that this quantity is comparable to the naive lower bound $\hat{R}^{\text{poly}}_{\alpha,S,P,\delta} \geq \frac{\alpha \delta^P + (1-\delta^P)}{P}$ obtained via minimizing the numerator ($S = P - 1$) and maximizing the denominator ($S = 0$) of $\hat{R}^{\text{poly}}_{\alpha,S,P,\delta}$. This intuitively suggests that the DIP coding scheme can potentially perform better than the polynomial coding scheme (Fig. 9 ascertains our intuition).

### 4.4.2 Constrained delay ($T < \infty$)

In this setting, each job-$i$, $i \in [1 : J]$, will be initiated in round-$i$ and has to be finished before the end of round-$(i + T)$. Recall from Algorithm 1 that for $t \in [1 : J + T]$, $\gamma'_t$ denotes the number of remaining mini-task results corresponding to job-$(t - T)$ that need to be obtained during round-$t$, in order to finish job-$(t - T)$ in round-$t$. As job-$i'$ is trivial whenever $i' \notin [1 : J]$, we have $\gamma_{i'+T} \triangleq 0$. Let $\tilde{C}(t)$ denote the number of workers who return results of all the mini-tasks assigned to them in round-$t$. Note that distribution of $\tilde{C}(t)$ will be different from that of $C(t)$ defined in (7), as here, the master can potentially wait for more workers to return all their mini-task results in round-$t$ as job-$(t - T)$ has to be finished by the end of round-$t$. Hence, we have $\tilde{C}(t) \geq C(t)$. We have:

$$\gamma'_t = \max\left(0, \beta - \sum_{j=t-T}^{t-1} \tilde{C}(j)\right), \tag{14}$$

where $\beta \triangleq xy$. Let $q_t$ denote the number of jobs in the range $[t - T : t]$ which have less than $\beta$ corresponding mini-tasks processed in the beginning of round-$t$. Let $\ell_t$ denote the number of non-trivial mini-tasks assigned to each worker in round-$t$. If master has already received $\beta$ or more mini-task results corresponding to job-$(t - T)$ before round-$t$, we have $\ell_t = q_t$. Otherwise, we have $\ell_t = q_t - 1 + \left\lceil \frac{\gamma'_t}{\lambda} \right\rceil$. It is straightforward to note that the following inequality holds in either case:

$$\ell_t \leq q_t + \frac{\gamma'_t}{\lambda}. \tag{15}$$

We have:

$$
\begin{aligned}
\mathbb{E}[\gamma'_t] &= \mathbb{E}\left[\max\left(0, \beta - \sum_{j=t-T}^{t-1} \tilde{C}(j)\right)\right] \\
&\leq \mathbb{E}\left[\max\left(0, \beta - \sum_{j=t-T}^{t-1} C(j)\right)\right] \\
&= \sum_{k=0}^{\beta} k.\mathbb{P}\left[\sum_{j=t-T}^{t-1} C(t) = \beta - k\right] \\
&\triangleq L_{T,\beta,\delta,P}.
\end{aligned}
\tag{16}
$$

Master waits for the stragglers to return their mini-task results in round-$t$ *only if* $\gamma'_t > 0$ and there are more than $P - \lambda$ stragglers in round-$t$. Let $p_{\lambda,P,\delta}(t)$ denote the probability of master waiting for a straggler in round-$t$. Thus, we have:

$$p_{\lambda,P,\delta}(t) \leq \sum_{j=0}^{\lambda-1} \binom{P}{j}(1 - \delta)^j \delta^{P-j} \triangleq p^*_{\lambda,P,\delta}. \tag{17}$$

The duration $r_t$ of round-$t$ clearly satisfies:

$$
r_t = \begin{cases} \frac{\ell_t \tau}{\beta}, & w.p. \quad 1 - p_{\lambda,P,\delta}(t), \\ \frac{\alpha \ell_t \tau}{\beta}, & w.p. \quad p_{\lambda,P,\delta}(t). \end{cases}
$$

Figure 9: A plot of $\delta$ vs. $\hat{R}$. In Fig. (a) and Fig. (b), we consider $\{\alpha = 5, P = 4, T = W - 1 = 3\}$ and $\{\alpha = 10, P = 6, T = W - 1 = 4\}$, respectively. For given $\{\alpha, P, \delta\}$, we minimize each of $\hat{R}^{\text{poly}}_{\alpha,S,P,\delta}$, $\hat{R}^{\text{DIP}}_{\alpha,\beta,\infty,\infty,P,\delta}$, $\hat{R}^{\text{DIP}}_{\alpha,\beta,T,\lambda,P,\delta}$, $\hat{R}^{\text{IDIP}}_{\alpha,N,W,P,\delta}$ (upper bound) with respect to $\{S \geq 1\}$, $\{\beta \geq P\}$, $\{\beta \geq P, \lambda\}$ and $\{N \geq 1\}$, respectively.

The expected processing time for $(J + T)$ rounds is now given by:

$$
\begin{aligned}
\mathbb{E}[R_J] &= \sum_{t=1}^{J+T} \mathbb{E}[r_t] = \sum_{t=1}^{J+T} \mathbb{E}[\frac{\ell_t \tau}{\beta}(\alpha p_{\lambda,P,\delta}(t) + (1 - p_{\lambda,P,\delta}(t)))] \\
&\overset{(a)}{\leq} \frac{\alpha p^*_{\lambda,P,\delta} + (1 - p^*_{\lambda,P,\delta})}{\beta} \tau \sum_{t=1}^{J+T} \left( \mathbb{E}[q_t] + \mathbb{E}[\frac{\gamma'_t}{\lambda}] \right) \\
&\overset{(b)}{\leq} \frac{\alpha p^*_{\lambda,P,\delta} + (1 - p^*_{\lambda,P,\delta})}{\beta} \tau \Big( \sum_{t=1}^{J+T} \mathbb{E}[q_t] + (J + T)\frac{L_{T,\beta,\delta,P}}{\lambda} \Big),
\end{aligned}
\tag{18}
$$

where $(a)$ follows from (15) and (17), $(b)$ follows from (16). Let $t'_j$ denote the number of rounds required to complete job-$j$ ($j \in [1 : J]$). It can be easily verified that:

$$
\sum_{j=1}^{J} t'_j = \sum_{t=1}^{J+T} q_t.
\tag{19}
$$

Using (18) and (19) we have:

$$
\mathbb{E}[R_J] \leq \frac{\alpha p^*_{\lambda,P,\delta} + (1 - p^*_{\lambda,P,\delta})}{\beta} \tau \Big( \sum_{j=1}^{J} \mathbb{E}[t'_j] + (J + T)\frac{L_{T,\beta,\delta,P}}{\lambda} \Big).
\tag{20}
$$

Note that in this scheme, compared to the unconstrained delay scenario, each job has to be finished with a delay of at most $T$ rounds. Hence, we have:

$$
\mathbb{E}[t'_j] \leq \min(T + 1, f_{\beta,\delta,P}),
\tag{21}
$$

where $f_{\beta,\delta,P}$ is as given by (10). Substituting (21) in (20), we have:

$$
\mathbb{E}[R_J] \leq \frac{\alpha p^*_{\lambda,P,\delta} + (1 - p^*_{\lambda,P,\delta})}{\beta} J\tau \left( \min(T + 1, f_{\beta,\delta,P}) + (1 + \frac{T}{J})\frac{L_{T,\beta,\delta,P}}{\lambda} \right),
$$

and consequently:

$$
\hat{R}^{\text{DIP}}_{\alpha,\beta,T,\lambda,P,\delta} \leq \frac{\alpha p^*_{\lambda,P,\delta} + (1 - p^*_{\lambda,P,\delta})}{\beta} \left( \min(T + 1, f_{\beta,\delta,P}) + \frac{L_{T,\beta,\delta,P}}{\lambda} \right).
$$

Figure 10: In Fig. (a), we show the transition probabilities of the Fritchman straggler model. In Fig. (b), we show a histogram of burst lengths seen by the model ($P_G = 0.2$, $P_B = 0.5$).

## 5 Experimental results

In this section, we evaluate the performance of proposed schemes by training $5$ neural network (NN) models concurrently (with learning rates $\{0.1, 0.15, 0.2, 0.25, 0.3\}$). Algorithms are implemented using mpi4py [19] and NumPy on a local university testbed. We use four virtual machines with 8GB of RAM and 4 vCPUs as workers and one more machine with 16GB of RAM and 8 vCPUs as the master. The master distributes jobs, keeps track of stragglers, and decodes the job results. During the wait-time to obtain results from workers in each round, master performs decoding of results from previous rounds and any necessary encoding for the next round. Master performs encoding and decoding at the same time on multiple vCPUs. Our experiments show that average encoding/decoding times are smaller than average processing times (denoted by $R \triangleq \frac{R_J}{J}$). Consequently, the effect encoding/decoding times have in our experiments is not significant.

Each NN model is fully connected with two hidden layers of size $3000$ followed by a ReLU activation. Training is performed for $250$ iterations over MNIST dataset with a batch size of $1024$ using SGD. We break down each iteration of training into $8$ sequential matrix-matrix multiplication jobs (job loads vary across these 8 jobs); 3 and 5 jobs respectively for forward and backward passes. The jobs belonging to the 5 NN models are interleaved so that job-$i$ belongs to model $((i-1) \bmod 5) + 1$. Hence, input matrices for job-$i$ is dependent on the result of job-$(i-5)$. It takes $T + 1$ rounds (worst-case) to deliver mini-task results corresponding to each job and then, one additional round for decoding. Thus, $T$ is set to 3. For the 5 NN models, we have in total $J = 250 * 5 * 8 = 10000$ jobs.

We run the experiments based on the i.i.d. straggler model, as well as the Fritchman model, which models presence of stragglers in bursts. In order to simulate stragglers, we make the to-be-stragglers perform their tasks $\alpha = 5$ times. We select best-performing code parameters for each straggler model using a simplified first order simulation. For the i.i.d. model, we set the straggler probability $\delta = 0.3$. The code parameters used are (i) polynomial: $\{S = 2, x = 2, y = 1\}$ (ii) DIP: $\{x = 2, y = 3, T = 3, \lambda = 1, \mu = 0.25\}$ (iii) IDIP: $\{x = 2, y = 3, T = 3, N = 10, W = 4, \mu = 0.25\}$ (the IDIP scheme here is the arbitrary straggler variant). For the Fritchman model, we consider three states (see Fig. 10), where a worker will be a straggler in round-$t$ iff it is in either state $B_1$ or $B_2$. The state transitions happen to every worker in the beginning of each round, with transition probabilities $P_G = 0.2$, $P_B = 0.5$. The code parameters used here are (i) polynomial: $\{S = 2, x = 2, y = 1\}$ (ii) DIP: $\{x = 2, y = 2, T = 3, \lambda = 2, \mu = 0.25\}$ (iii) IDIP: $\{x = 2, y = 6, T = 3, B = 2, \epsilon = 2, W = 3, \mu = 0.25\}$ (the IDIP scheme here is the bursty straggler variant). A performance summary is provided in Table 1. Fig. 11 clearly depicts the improvement newly proposed schemes provide over the polynomial coding scheme. DIP, IDIP schemes register reductions of $36\%$ ($32\%$) and $24\%$ ($41\%$), respectively, in the average processing time ($R$) over polynomial codes, for i.i.d. (Fritchman) straggler model. Note that experimental results are in agreement with the numerical results, which assume the i.i.d. straggler model and predict superior performance of DIP scheme over both IDIP and polynomial schemes. For the Fritchman model, IDIP emerges to be the best-performing scheme. The complete set of codes used for these experiments, are available at: `https://github.com/erfanhss/SequentialCodedComputing`.

(a) i.i.d. straggler model   (b) i.i.d. straggler model

(c) Fritchman model   (d) Fritchman model

Figure 11: Experimental results for training neural networks. Fig. (a) & (c): round vs. cumulative processing time; Fig. (b) & (d): time vs. test loss for NN model-1.

(a) i.i.d. straggler model   (b) Fritchman model

Figure 12: Experimental results for training neural networks. Round (first 200) vs. task load. The task loads vary across rounds for all schemes, as job loads are not the same across rounds. In particular, the load-variation can be seen to be amplified for the DIP scheme (due to its inherent variable load nature) in Fig. (b).

Table 1: Performance summary for two straggler models, in terms of encoding/decoding times and $R$.

<table>
<tr><td colspan="4" align="center">(a) i.i.d. model</td><td colspan="4" align="center">(b) Fritchman model</td></tr>
<tr><td>Scheme</td><td>Enc. time</td><td>Dec. time</td><td>$R$</td><td>Scheme</td><td>Enc. time</td><td>Dec. time</td><td>$R$</td></tr>
<tr><td>Uncoded</td><td>0</td><td>0</td><td>0.3</td><td>Uncoded</td><td>0</td><td>0</td><td>0.574</td></tr>
<tr><td>Poly</td><td>0.076</td><td>0.077</td><td>0.152</td><td>Poly</td><td>0.045</td><td>0.187</td><td>0.482</td></tr>
<tr><td>DIP</td><td>0.091</td><td>0.082</td><td>0.096</td><td>DIP</td><td>0.065</td><td>0.206</td><td>0.327</td></tr>
<tr><td>IDIP</td><td>0.052</td><td>0.078</td><td>0.115</td><td>IDIP</td><td>0.024</td><td>0.193</td><td>0.281</td></tr>
</table>

**Remark 5.1** (Applicability of the scheme in general). Even though we discuss the specific application of training multiple NN simultaneously, our framework suits well in any situation where the master is interested in finishing quickly a collection of multiple independent sequences of matrix multiplications (dependencies are permitted within a sequence). For instance, solving multiple systems of linear equations through an iterative algorithm such as the Jacobi method.

## A   Proof of Theorem 3.1

We assume that either $m \leq n$ or $p \leq n$ (the authors of [3] also make the same assumption), so that the matrix product $X(t) * Y(t)$ is not trivially rank-deficient (which degenerates the problem). We are interested in the exact recovery of the product $X(t) * Y(t)$, which can take any value in $\mathbb{R}^{m \times p}$ (say $b$ bits for each real value, owing to the finite precision). Thus, finishing each job-$t$, i.e., computing all the entries of $X(t) * Y(t)$, requires master to have access to at least $mpb$ bits.

Let $N = N_1 W + N_2$, where $0 \leq N_2 < W$. Consider the periodic straggler pattern depicted in Fig. 13, which conforms to the $(N, W)$-arbitrary straggler model. Let $L$ denote the maximum of normalized task loads across all rounds $[1 : J + T]$. Note that $L$ is of the form $\frac{\ell'}{xy}$, where $\ell'$ is the number of mini-tasks assigned to each worker for the round with the maximum normalized task load. Each mini-task result is a matrix $\in \mathbb{R}^{\frac{m}{x} \times \frac{p}{y}}$, which contains $\frac{mp}{xy} b$ bits. Hence, a worker with $\ell_t \leq \ell'$ mini-tasks in round-$t$ can transfer at most $\ell' \frac{mp}{xy} b = Lmpb$ bits to the master, when it finishes all the mini-tasks assigned to it.

For some $i \geq 1$, consider the first $iW$ rounds. A total of $iW$ jobs are initiated in these rounds. In order to finish this many jobs, the master needs access to at least $iW mpb$ bits. The total number of bits returned by the workers in these $iW$ rounds is at most $i(PW - N)Lmpb$ (as stragglers are assumed to provide no results as per the straggler model). The amount of pending information which is required to finish all these $iW$ jobs is thus at least $\max(\{iW - iPWL + iNL, 0\})mpb = i(PW - N)\max(\{L^* - L, 0\})mpb$ bits. Assume that $L < L^*$. Even if there are no stragglers from round-$(iW + 1)$ onwards, the master will still take at least $\left\lceil \frac{i(PW-N)(L^*-L)}{PL} \right\rceil$ more rounds to accumulate $i(PW - N)(L^* - L)mpb$ bits. As it scales with $i$, for sufficiently large $i$, we have $\left\lceil \frac{i(PW-N)(L^*-L)}{PL} \right\rceil > T$ and the delay constraint $T$ is not met. Thus, our assumption does not hold and $L \geq L^*$.

## B   Proof of Theorem 3.2

Consider the straggler pattern depicted in Fig. 14, which conforms to the $(B, \epsilon, W)$-bursty straggler model. Each row in the figure corresponds to a worker, whereas each column corresponds to a round. First $\epsilon$ workers are experiencing a periodic pattern of $B$ consecutive straggling rounds followed by a guard interval of $W - 1$ rounds. Let $L$ denote the maximum of normalized task loads across all rounds. For some $i \geq 1$, consider the first $i(B + W - 1)$ rounds. A total of $i(B + W - 1)$ jobs are initiated in these rounds. The total number of bits returned by the workers in these $i(B + W - 1)$ rounds is $i(P(B + W - 1) - \epsilon B)Lmpb$ (as stragglers are assumed to provide no results as per the model). The amount of pending information required by the master to finish all these $i(B + W - 1)$ jobs is thus at least $\max(\{i(B+W-1) - iP(B+W-1)L + iB\epsilon L, 0\})mpb = i(P(B+W-1) - B\epsilon)\max(\{L^* - L, 0\})mpb$ bits. Assume that $L < L^*$. Even if there are no stragglers from round-$(i(B + W - 1) + 1)$ onwards, the master will still require at least $\left\lceil \frac{i(P(B+W-1)-B\epsilon)(L^*-L)}{PL} \right\rceil$ more rounds to access these

Figure 13: A periodic straggler pattern conforming to the $(N, W)$-arbitrary straggler model.

Figure 14: A periodic straggler pattern conforming to the $(B, \epsilon, W)$-bursty straggler model.

many bits. As it scales with $i$, for sufficiently large $i$, we have $\left\lceil \frac{i(P(B+W-1)-B\epsilon)(L^*-L)}{PL} \right\rceil > T$. Hence, the delay constraint $T$ is clearly not met. Thus, $L \geq L^*$ when $J \to \infty$ and $T < \infty$. For the special case of $B = W$, consider the straggler pattern shown in Fig. 15. Following similar arguments as above, it can be inferred that $L \geq L^*$.

## C   Additional experiments with constant job load

In this section, we describe results of some additional experiments we conducted to evaluate the performance of our schemes, when job loads are of constant size. We take the total number of jobs ($J$) to be 300. All the matrices $\{X(i)\}, \{Y(i)\}$ have dimension $4000 \times 4000$. We use here the same experimental setup as in Sec. 5 with four virtual machines. Stragglers are produced based on the Gilbert-Elliott (GE) model. GE model is nothing but a Fritchman model with two states (see Fig. 16). A worker here will be a straggler in round-$t$ iff it is in state $B$. The state transitions happen to every worker in the beginning of each round, with transition probabilities $P_G = P_B = 0.2$. The code parameters used here are (i) polynomial: $\{S = 2, x = 2, y = 1\}$ (ii) DIP: $\{x = 2, y = 2, T = 4, \lambda = 3, \mu = 0.25\}$ (iii) IDIP: $\{x = 2, y = 7, T = 4, B = 2, \epsilon = 3, W = 4, \mu = 0.25\}$ (the IDIP scheme here is the bursty straggler variant). A performance summary is provided in Table 2. In terms of average processing time, DIP and IDIP schemes achieve reductions of 57% and 62%, respectively, over polynomial codes.

Figure 15: A straggler pattern conforming to the $(B, \epsilon, W)$-bursty straggler model, when $B = W$.

Figure 16: Transition probabilities of the Gilbert-Elliott straggler model.

Table 2: Performance summary in terms of encoding/decoding times and $R$.

| Scheme | Enc. time | Dec. time | $R$ |
|--------|-----------|-----------|-----|
| Uncoded | 0 | 0 | 9.33 |
| Poly | 0.32 | 1.36 | 8.40 |
| DIP | 0.38 | 2.45 | 3.60 |
| IDIP | 0.16 | 1.18 | 3.12 |