[Reviews · NeurIPS 2020]

Review 1

Summary and Contributions: This paper studies the problem of performing large scale matrix multiplications over a distributed computing setup in the presence of stragglers. In the naive scheme, one divides the matrix multiplication task to many mini-tasks (multiplication of sub-blocks of the underlying matrices) which are assigned to different worker nodes. In this case, the master node needs to wait for the slowest workers, i.e., stragglers, to obtain the final result. Recently, coding for computation has been explored to counter the stragglers. Here, one redundantly assign the mini-tasks to the workers and the master can obtain the final result as long as enough number of worker complete their mini-tasks in a timely manner. For distribution matrix multiplication, optimal coding schemes are known in the literature (e.g., polynomial coding). However, this coding scheme focus on one large scale matrix multiplication at a time. Thus, the coding can only be performed across workers. In this paper, the authors consider the problem of sequential matrix multiplication where the objective is to perform multiple large scale matrix multiplications such that the matrix multiplication task arriving at round t must be completed by round t + T. The additional freedom associated with T>0 allows the authors to perform encoding both across workers and time. For this novel problem of coded sequential matrix multiplication, the paper proposes two coding schemes that generalize polynomial codes. The paper then analyzes the completion time of these coding schemes for a simple iid straggler model and shows the advantages over polynomial coding or naive uncoded approach. The paper also evaluates the proposed scheme on a real distributed computing setup where sequential matrix multiplications correspond to computations performed during (concurrent) iterative training of independent neural networks on MNIST. The paper shows that the average processing time of the proposed schemes is better than that of polynomial coding and uncoded approach two stochastic models for stragglers: 1) iid model and 2) Fritchman model.

Strengths: The paper introduced the problem of coded sequential matrix multiplication in the presence of stragglers. The paper proposes two coding schemes for the underlying problem where underlying matrices are encoded both across workers and across time. This enables the system to tolerate stragglers that can appear in a burst as long as the total number of stragglers in a given time window is bounded. For stochastic straggler models, the paper show, both analytically and experimentally, that the proposed coding schemes lead to lower processing time (time required to complete the matrix multiplications) as compared to the existing baselines.

Weaknesses: Although addressing a novel problem, the proposed coding schemes in the paper do not introduce any novel coding techniques. Thus, it becomes even more important than the paper carries out a thorough empirical evaluation of the coding scheme. The paper, in its current form, is light on experimental evaluation. How realistic are the stochastic models for the stragglers considered in the paper in the context of large scale distributed computation.

Correctness: As far as the reviewer can tell, the results in the paper are correct.

Clarity: The main body of the paper glosses over many important details that are presented in the supplementary material. The supplementary appears as a complete paper in itself. The reader cannot be expected to read the entire paper while referring to the supplementary. Please try to include important details such as motivation for the coding across time, explanation of the proposed coding schemes in the main body. When referring the reader to the supplementary, please refer to the specific sections of the supplementary.

Relation to Prior Work: The paper adequately discusses the relevant prior work on coded computation. The sequential matrix multiplication problem where one requires to complete each arriving matrix multiplication task within a pre-specified time range has some similarity with the problem of coding for streaming. Could the authors comment on this?

Reproducibility: Yes

Additional Feedback: In line 215, please add a citation regarding the condition number of Vandermonde matrices. The paper separately encoded each job. Could the authors comment on the utility of the coding schemes that perform encoding across multiple jobs? ###################### Post author response phase ################## Thank you for taking the time to respond to my comments. As acknowledged in my review, the paper does propose a novel problem setting. However, the experimental results are a bit limited. Also, there is scope for enhancing the structure of the paper with more streamlined referencing to the supplementary. After going through the authors' responses and other reviewers' comments, I have decided to leave my score unchanged. Please include a discussion on the connections/differences between your work and coding for streaming in the revised version.


Review 2

Summary and Contributions: This paper considers the problem of coded computation for matrix multiplication. Different from earlier works that have addressed the problem, it proposes a novel way of looking at a sequence of matrix multiplications through a "temporal" dimension. This dimension indicates the number of steps of coordination between master-workers before which the result of a particular matrix multiplication in the sequence is expected. This problem is termed Coded sequential multiplication in this paper. For this novel view, the work proposes extensions of the polynomial codes of [3] taking the temporal aspect into account. To analyze the efficiency of the proposed methods, the paper also extends the arbitrary straggler model to multiple rounds, called (N,W) straggler model. The asymptotic load of the proposed methods is analyzed under the model, as well as for polynomial codes, and it is shown to be better. Simulations to train multiple simultaneous NN are also run, as a particular application of coded sequential multiplication.

Strengths: The proposed view on coded matrix multiplication is very novel and would be of interest to the NeurIPS community. The paper is also fairly well-written. Despite being notationally heavy, the authors have done a good job of repeating notational details at the right spots in the paper, making it easy to read. The authors also do a good job of motivating why incorporating the temporal dimension can provide more gains over one-shot coded matrix multiplication.

Weaknesses: The practical application suggested of training multiple NN simultaneously, seems somewhat limited.

Correctness: Yes, the results seem correct. Though, I have not verified the proofs in the supplementary material in detail.

Clarity: Yes

Relation to Prior Work: Yes

Reproducibility: Yes

Additional Feedback: - In lines 203,204, can you please clarify what 'u' is ? - In line 235, "We consider in this section, a probabilistic, i.i.d. straggler model, i.e., any worker in any given round will be a straggler with probability \delta", can you explain the relation between this model and the (N,W) straggler model. It seems that they co-exist ? - In line 259, "Through numerical evaluation in Fig. 1, it can be observed that DIP scheme outperforms polynomial and IDIP schemes". Please explain why. It was my impression that IDIP should outperform DIP under the (N,W) straggler model. #### After author response #### Thank you for responding to my comments. My score remains unchanged, though I also do agree with the other reviews regarding limited experimental evaluation. Please mention at least a discussion around other applications of this problem in the revision.


Review 3

Summary and Contributions: This paper studies a generalized version of coded matrix multiplications in multiple rounds. The main observation is that when extending the computation to multiple rounds, the worst-case straggler tolerance improves due to the ability to launch delayed recomputation of failed subtasks (thus the stragglers become more evenly distributed across time). The authors show in both theory and experiments that the coded computing schemes based on this idea can achieve better cumulative processing time compared to the static scheme [3]

Strengths: * The problem formulation does genenerlize the conventional setting in coding for straggler mitigation * The authors provide optimality proofs of the proposed schemes under a well-defined problem formulation * The authors show some improvement over [3]

Weaknesses: * The improvement seems limited * The experiment setting is semi-simulated because this not how people train neural networks (i.e., partition the computation into sequential matrix multiplications and distribute each partial job to several workers) * The studied problem seems to be a classical extension under the framework of delay-vs-throughput tradeoff, which limits the novelty

Correctness: The claims sound correct to me. The authors did provide a thorough analysis on the proposed schemes

Clarity: The paper writing is ok. Sometimes I have to go to the supplementary materials in order to understand the details.

Relation to Prior Work: There is not much discussion on the connection to and difference from prior works

Reproducibility: Yes

Additional Feedback: (After rebuttal) I want to thank the authors for addressing my comments. I have read the authors' rebuttal and the other reviewers' comments and have carefully rethought the paper, including its connections to other related works. Since the authors have thoroughly addressed two of my three major concerns given the limited rebuttal space, I want to increase my overall score from 5 to 6. The reason that my final score is still a borderline score is mainly because of the unaddressed first comment and my third comment regarding novelty. Although I agree with the authors that the proposed scheme, especially the idea of actively selecting uncoded matrix-multiplication, has some novelty, the idea of amortizing tasks over time is a quite commonly explored theme in communication networks. But I agree that this idea has not been fully explored in coded computing. Regarding the experiments, I hope the authors can find and demonstrate applications that are motivated by practical implementation such as those proposed in the rebuttal. So maybe conducting well-developed experiments and improving the empirical evaluations can be the primary focus to improve the paper.

[Author Response · NeurIPS 2020]

We thank the reviewers for carefully reading the manuscript and providing their comments.

**R1+R3: Novelty of results.** As the first work to our knowledge that applies coding along the temporal and "spatial" dimensions for distributed Matrix Multiplication (MM), we believe the paper could spark interest in follow-up theoretical and experimental investigations. All previous works (e.g. Polynomial Codes (PC's)) have focused on one-shot MM that involves coding only across workers (i.e., spatial dimension). The advantages of our framework are two-fold (1) by exploiting the temporal dimension, our proposed schemes tolerate more straggler patterns than PC under the same normalized load at the workers. In particular, if PC tolerates $S$ stragglers in each round, then under the same worker load, our proposed schemes tolerate a total of $(T+1) \cdot S$ stragglers in each sliding window of $T+1$ consecutive rounds, with delay $T$ (please see "Motivating example" in Sec. 2.1 of supplementary material). (2) Our schemes exploit the feedback of previously occurred straggler patterns from previous rounds to adapt the computations to be performed. In particular, in the IDIP scheme, the master judiciously opts for "uncoded mini-tasks" wherever possible, based on such feedback, with the objective to reduce overall decoding complexity. Previous works did not explore the temporal dimension and hence, did not have access to this feedback. The other theoretical results in the paper include (i) showing the optimality of DIP/IDIP schemes under the $(N, W)$-Straggler Model (SM) and (ii) an analytical comparison of the expected run-time of various schemes (which illustrates advantage of our schemes) under an i.i.d. SM.

**R1: On the choice of stochastic SM's** We emphasize that although the performance analysis of our coding schemes assumes certain stochastic models, our coding schemes can be applied irrespective of the actual pattern of stragglers. Please refer to the discussion on page 5 (last paragraph) in main paper. Thus, our coding schemes are not limited by the assumed stochastic model. The i.i.d. SM in Section 4 enables us to develop analytical performance bounds and develop insights. The use of Fritchman model in Section 5 is motivated by its ability to model occurrence of bursty stragglers. In [13], the authors note that speed variation in an Amazon EC2 credit-based instance can be closely modeled by a similar bursty stochastic model. Finally, we note that our claim in Section 4.3 of the supplementary material that $p^{\text{IDIP}} \leq p^{\text{poly}}$, i.e., the master is more likely to require straggler nodes to complete their jobs in the PC scheme, does not rely on i.i.d. model, but instead holds for any stochastic model. This implies that coding across time dimension always improves upon one-shot PC, regardless of the SM.

**R1: Paper is light on experimental evaluation** Experimental results in Sec. 5 report encoding/decoding/processing times measured on workers and master, when training an NN. Consistent with prior work, we artificially injected stragglers during training to develop insights into performance. E.g., in Fig 5(f), we demonstrate that when bursty stragglers are introduced, the instantaneous load of IDIP scheme remains consistently lower than PC, while the DIP scheme requires high load. Thus, the proposed IDIP scheme could have significant impact in practice when bursty stragglers are reported. We note that the contribution of the paper is to propose new coding schemes along with optimality guarantees, analysis for i.i.d. SM as well as experimental evaluation to develop insights into the performance. A large scale experimental study, while interesting, is beyond the scope of the present paper.

**R2+R3: Applicability of the scheme in general** While we focus on training multiple NN simultaneously in the paper, the framework suits well in any application where the master is interested in finishing quickly a collection of multiple independent sequences of MM's (dependencies are permitted within a sequence). Clearly, this is applicable if one is interested in solving multiple systems of linear equations through an iterative algorithm such as Jacobi method. Cloud platforms providing route planning, page ranking services solve multiple systems of equations in every second.

**R1+R3: Comparison with Streaming Codes (SC's)** While we are aware of the literature on SC's, there are fundamental differences in the two approaches, because of which, SC constructions do not seem to be applicable to MM problems. For instance, consider the SC toy example, where packets $p_1$, $p_2$, $p_1 + p_2$ are transmitted in rounds $1, 2, 3$ (can recover any lost packet with delay 2). Extending this to our setting, suppose a worker computes $A_1 x_1$, $A_2 x_2$ and $A_1 x_1 + A_2 x_2$ in successive rounds. This scheme is sub-optimal as $A_1 x_1 + A_2 x_2$ involves 2 matrix-vector multiplications.

**R1: Additional comments** Coding across jobs may not be possible as matrices in different jobs may have incompatible dimensions or data. Moreover, for the (N,W)-SM, our proposed schemes are already optimal.

**R2: Additional comments** (1) $u_i$ (subscript was missing) indicates #uncoded mini-tasks of job-$i$. (2) We missed to mention that i.i.d. SM is a stochastic extension of the deterministic $(N, W)$-SM. (3) Under the $(N, W)$-SM, IDIP scheme does perform better than DIP. However, in Fig. 1, the performance is plotted w.r.t. i.i.d. SM. DIP scheme is allowed to increase instantaneous worker load in future rounds to handle failed mini-tasks, whereas IDIP scheme waits for stragglers to complete their mini-tasks (round duration expands by $\alpha$) when non-ideal straggler patterns are encountered. Our numerical simulations indicate that under i.i.d. SM, for $\alpha >> 1$, IDIP scheme gets penalized more.

**R3: Connections to delay-vs-throughput tradeoff, prior work** Apart from streaming codes, we do not see any connection between ours and the delay vs. throughput tradeoff framework for communication networks. While we believe we have already addressed in the paper how our coding approach differs from the existing one-shot approaches, we will include more details in the revision to bring out further, the differences and motivation for our approach.

[Meta-Review · NeurIPS 2020]

Reviewers agree that this paper addresses an interesting (and potentially very important) problem. The method has certain novelty in the context of coded computing. The experiments could be more thorough and more realistic. The paper is a bit hard to read and some details could be better explained. For example, the authors refer to [3] for important steps which might not be familiar to general audience. Computing the inverse involving the Vandermonde matrix could potentially lead to numerical problems, which is not elaborated in the paper.,